# Faith and Fate:
# Limits of Transformers on Compositionality

**Nouha Dziri**[1*], **Ximing Lu**[1,2*], **Melanie Sclar**[2*],
**Xiang Lorraine Li**[1†], **Liwei Jiang**[1,2†], **Bill Yuchen Lin**[1†],
**Peter West**[1,2], **Chandra Bhagavatula**[1], **Ronan Le Bras**[1], **Jena D. Hwang**[1], **Soumya Sanyal**[3],
**Sean Welleck**[1,2], **Xiang Ren**[1,3], **Allyson Ettinger**[1,4], **Zaid Harchaoui**[1,2], **Yejin Choi**[1,2]

[1]Allen Institute for Artificial Intelligence    [2]University of Washington
[3]University of Southern California    [4]University of Chicago

nouhad@allenai.org, ximinglu@allenai.org, msclar@cs.washington.edu

## Abstract

Transformer large language models (LLMs) have sparked admiration for their exceptional performance on tasks that demand intricate multi-step reasoning. Yet, these models simultaneously show failures on surprisingly trivial problems. This begs the question: Are these errors incidental, or do they signal more substantial limitations? In an attempt to demystify transformer LLMs, we investigate the limits of these models across three representative *compositional* tasks—multi-digit multiplication, logic grid puzzles, and a classic dynamic programming problem. These tasks require breaking problems down into sub-steps and synthesizing these steps into a precise answer. We formulate compositional tasks as computation graphs to systematically quantify the level of complexity, and break down reasoning steps into intermediate sub-procedures. Our empirical findings suggest that transformer LLMs solve compositional tasks by reducing multi-step compositional reasoning into linearized subgraph matching, without necessarily developing systematic problem-solving skills. To round off our empirical study, we provide theoretical arguments on abstract multi-step reasoning problems that highlight how autoregressive generations' performance can rapidly decay with increased task complexity.

## 1 Introduction

*"It was the epoch of belief, it was the epoch of incredulity." – Charles Dickens, A Tale of Two Cities*

Large-scale transformers such as ChatGPT [57] and GPT4 [58] demonstrate unprecedented capabilities [57, 74, 11, 15, 85], even noted as "sparks of AGI" [12]. In stark contrast, the same models sometimes struggle with simple, intuitive tasks [9, 62, 40]. For instance, humans can solve 3-digit by 3-digit multiplication arithmetic after learning basic calculation rules [22, 34]. Yet, off-the-shelf ChatGPT and GPT4 achieve only 55% and 59% accuracies on this task, respectively (§3).

The striking discrepancy between the impressive successes of transformer LLMs on *seemingly complex* tasks and the astonishing failures on *seemingly trivial* tasks spark critical open questions about how to faithfully interpret their mixed capabilities. Under what conditions do transformers succeed, fail, and why? What types of errors do they make? Can transformers uncover implicit problem-solving rules or be taught to follow reasoning paths?

Seeking thorough answers to these questions remains an open research challenge. However, we offer novel insights into the fundamental limits of transformers[2], centered around *compositional*

---

* First co-authors.    † Second co-authors.

[2]For brevity, we use 'transformers' to refer to 'autoregressive transformer LLMs' throughout the paper.

37th Conference on Neural Information Processing Systems (NeurIPS 2023).

*problems* that require strict multi-hop reasoning to derive correct predictions. Applying step-by-step reasoning is fundamental to human intelligence [69, 68]. These compositional problems present compelling challenges for AI systems as they require combining basic reasoning operations to follow computational paths that arrive at unique correct solutions. In particular, we study three straightforward and flexible representative compositional tasks: long-form multiplication, logic grid puzzles (i.e., Einstein's puzzle [61]), and a classic dynamic programming problem.

We propose two hypotheses. **First**, transformers solve compositional tasks by reducing multi-step compositional reasoning into linearized path matching. This contrasts with the systematic multi-step reasoning approach that learns to apply underlying *computational rules* required for building correct answers [71, 37, 27]. Shortcut learning [29] via pattern-matching may yield fast correct answers when similar compositional patterns are available during training but does not allow for robust generalization to uncommon or complex examples. **Second**, due to error propagation, transformers may have inherent limitations on solving high-complexity compositional tasks that exhibit novel patterns. Errors in the early stages of the computational process can lead to substantial compounding errors in subsequent steps, preventing models from finding correct solutions.

To investigate our hypotheses, we formulate compositional tasks as *computation graphs*. These graphs break down problem-solving into submodular functional steps, enabling structured measurements of complexity and verbalization of computational steps as input sequences to language models. Moreover, we leverage information gain to predict patterns that models are likely to learn based on the underlying task distribution without the need to perform full computations within the graph.

Empirical results show that training on task-specific data leads to near-perfect performance on in-domain instances and under low compositional complexity, but fails drastically on instances outside of this region. This substantial gap suggests that systematic problem-solving capabilities do not emerge from maximum likelihood training [5] on input-output sequences, even when prompted or trained with human-like reasoning steps (i.e., a linearization of computation graphs; §3.1). Models' success can be attributed, in part, to their exposure to training examples sub-graphs that involve the same computations required for solving test examples (see Section 3.2.2) In order to gain a deeper understanding of models' failures, we conduct a comprehensive analysis by decomposing their computation graphs and examining different error types. We find that while models can memorize single-step operations, they fail to compose them into correct reasoning paths, suggesting that they mostly make predictions based on shallow, rote learning rather than a deep, holistic task understanding (§3.2.3). Importantly, we provide theoretical evidence of exponential error accumulation using abstract compositional tasks. All tasks analyzed empirically in this paper are instantiations of these abstractions (§4). We argue that transformers could be inherently limited in solving compositionally complex tasks out-of-the-box[3].

As transformers continue to make tangible real-world impacts, it is pressing to interpret their remarkable performance critically. Our work takes a realistic look at the limitations of transformers in the context of compositional tasks. To shed light on practical future steps, we identify directions for addressing these limitations, such as using transformers for tasks that could be decomposed into few reasoning steps, tasks where evaluation may afford some leniency, and using transformers in combination with planning modules or refinement methods to improve their generations. To advance language AI, fundamental innovations are required to address or complement these limitations.

## 2 Measuring Limitations of Transformers in Compositional Tasks

Human problem-solving skills can be conceptualized as a graph structure, where each vertex represents a partial solution and the edges represent operators that can be applied to modify these solutions. As we will outline next and illustrate in Figure 1, we use computation graphs and corresponding metrics to methodically evaluate transformers' reasoning abilities.

### 2.1 Computation Graph Definition

Let $A$ be a deterministic algorithm (function), and let $\mathcal{F}_A$ be a set of primitives (functions) the algorithm uses in its execution. Assuming the inputs $\mathbf{x}$ to algorithm $A$ are given, we define $A(\mathbf{x})$'s static computation graph $G_{A(\mathbf{x})}$. $G_{A(\mathbf{x})} = (V, E, s, op)$ is a directed acyclic graph. Nodes $V$

---

[3]Code and data are available at `https://github.com/nouhadziri/faith-and-fate`

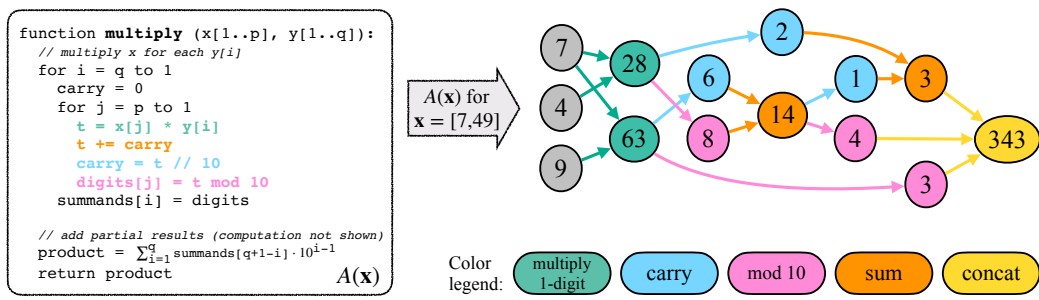

Figure 1: Transformation of an algorithm $A$ to its computational graph $G_{A(\mathbf{x})}$. The depicted example is of long-form multiplication algorithm $A$, for inputs $\mathbf{x} = [7, 49]$ (i.e. computing $7 \times 49$).

represent all variables' values during $A$'s execution: each node $v \in V$ has a value $s(v) \in \mathbb{R}$ associated. Edges $E$ represent the function arguments involved in some computation: for each non-source node $v \in V$, let $U = \{u_1, \ldots, u_j\} \subset V^j$ be its parent nodes. Then, $s(v) = f(u_1, \ldots, u_j)$ for some $f \in \mathcal{F}_A$. Since each node $v$ is uniquely defined by the computation of a single primitive $f$, we define $op : V \to \mathcal{F}_A$ as $op(v) = f$.

Let $S \subset V$ be the source nodes of $G_{A(\mathbf{x})}$ and without loss of generality, let $o \in V$ be its sole leaf node. By definition, $S \equiv \mathbf{x}$ and $A(\mathbf{x}) = s(o)$, representing the input and output of $A$ respectively.

To be able to train and evaluate a language model's ability to follow algorithm $A$ we must linearize $G_{A(\mathbf{x})}$. Since we only consider autoregressive models, this linearization must also be a topological ordering.

## 2.2 Quantifying Compositional Complexity using Graph Metrics

$A$'s representation as a computation graph $G_{A(\mathbf{x})}$ enables measuring task complexity from many angles.

We define a node $v \in V$'s *layer number* as the length of the longest path from a source node to $v$ in the directed acyclic graph $G_{A(\mathbf{c})}$. We then define the **reasoning depth** as the largest layer number in the graph. In computation graphs, reasoning depth is a proxy for the maximum level of multi-hop reasoning required to solve the task.

Let $d_S : V \to \mathbb{N}_0$ be the shortest distance to any of $G$'s source nodes $S \subset V$. We define the **reasoning width** of a graph as the mode of $\{d(v) : v \in V\}$. This metric aims to measure the maximum number of variables required to maintain in parallel during the computation. Relatedly, we also define the **average parallelism** of a graph as the ratio between $|V|$ and its reasoning depth. This aims to compute the average width in computation through the graph, and not just in its mode.

## 2.3 Predicting Surface Patterns through Relative Information Gain

When evaluating model performance, we may observe partially correct answers even in an overall incorrect response. To understand model strategies in these partial successes, we use Relative Information Gain to predict surface patterns that models are likely to recognize. We represent task $T$ as a distribution $(X_1, \ldots, X_n, Y_1, \ldots, Y_m)$ and measure the amount of (normalized) information gained about an output element $Y_j$ by observing a subset of input random variables $X \subset \{X_1, \ldots, X_n\}$:

$$\text{RelativeIG}(Y_j, X) = \frac{H(Y_j) - H(Y_j|X)}{H(Y_j)} \in [0, 1] \tag{1}$$

RelativeIG may be used to analyze the influence of any node in the computation graph (as defined in §2.1) with respect to a set of its ancestors; in particular, output nodes with respect to input nodes.

## 2.4 Exploring Three Representative Compositional Tasks: Definitions

**Multiplication** Multi-digit multiplication requires executing operations with numerical symbols based on procedural rules [34]. This task has multiple algorithmic solutions; in constructing computation graphs, we use the well-known $O(k_1 k_2)$ long-form multiplication algorithm for computing $x \cdot y$, where $x$ has $k_1 \leq 5$ digits and $y$ has $k_2 \leq 5$ digits in base 10. See §A.1 for data construction details.

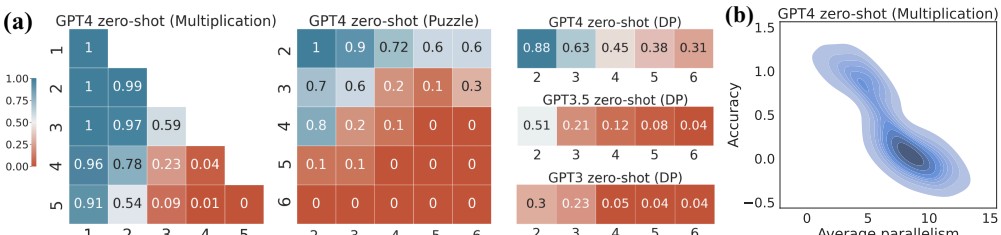

Figure 2: (a) **Zero-shot accuracy**. Axes refer to problem sizes (number of digits in multiplication, number of houses and attributes in puzzle, and sequence length in the DP task). Transformers' accuracy decreases to near zero as task complexity increases, measuring task complexity by the problem size. (b) **Average parallelism** negatively correlates with accuracy.

To instantiate $G_{A(\mathbf{x})}$, let $\mathcal{F}_A = \{$one-digit multiplication, sum, mod 10, carry over, concatenation$\}$. Source nodes $S$ are digits of input numbers, leaf node $o$ is the final output, and intermediate nodes $v$ are partial results generated during execution of the long-form multiplication algorithm (see Figure 1).

**Einstein's Puzzle**   Einstein's puzzle is a well-known logic puzzle often used as a benchmark for solving constraint satisfaction problems [61]. It involves a list of houses with different attributes (e.g., owner's name, pets), and the goal is to determine which attributes belong to each house by combining a set of pre-defined natural language clues or constraints. The solution to the puzzle is a matrix of size $K \times M$, where $K$ represents the number of houses and $M$ the number of attributes. As $K$ and $M$ increase, synthesizing different partial solutions that satisfy individual constraints becomes highly compositionally complex. To construct the computation graph, we consider a greedy algorithm that iteratively eliminates possible solutions by filling at least one cell each time. It deterministically fills the cell(s) that requires the minimum number of clues among all current unfilled cells. We refer to this as the *elimination function*. See §A.2 for examples, data construction, and algorithm details.

To instantiate $G_{A(\mathbf{x})}$, let $\mathcal{F}_A = \{$elimination function$\}$. The source nodes are the clues, all intermediate nodes are partially-filled matrices, and the output node is a fully-filled solution matrix.

**Dynamic Programming Problem**   Dynamic programming (DP) recursively breaks down complex problems into simpler sub-problems, so problems solved using this technique are compositional. We analyze a classic relaxation of the NP-complete Maximum Weighted Independent Set problem [39]: *Given a sequence of integers, find a subsequence with the highest sum, such that no two numbers in the subsequence are adjacent in the original sequence.* This relaxation may be solved in $O(n)$ time using DP. See the solution in §A.3. In the experiments, we restrict each integer to the $[-5, 5]$ range.

To instantiate $G_{A(\mathbf{x})}$, let $\mathcal{F}_A = \{$equals, and, not, indicator function, sum, max$\}$. Source nodes are elements of the input list, and the output node is a list that for each element indicates whether it should be selected. We select an $O(n)$ algorithm since $G_{A(\mathbf{x})}$'s size is proportional to $A$'s complexity.

## 3   Testing the Limits of Transformers: Empirical Evidence

**Experimental Setup**   To understand the capabilities of LLMs, we evaluate GPT3 (`text-davinci-003`) [11], ChatGPT (`GPT-3.5-turbo`) [57] and GPT4 (`gpt-4`) [58] using zero-shot, few-shot, and finetuning techniques. To enable the generation of computation graphs beyond the final answers, we use the concept of *scratchpads* [56]. Scratchpads are a verbalization of the computation graphs (i.e., a linearized representation of a topological ordering of $G_{A(\mathbf{x})}$). Overall, we consider *question-answer* and *question-scratchpad* formats for few-shot and finetuning settings to gauge models' capabilities for learning with and without explicit reasoning. See details of additional models and experimental configurations in §B and examples of scratchpad in §A.

### 3.1   Testing the Limits of Transformers with Zero-shot, Few-shot and Finetuning

**Limits of transformers in zero- and few-shot settings**   To investigate the inherent problem-solving capabilities of LLMs, we begin by analyzing models' zero-shot and few-shot performances on our compositional tasks. As shown in Figure 2, task performances deteriorate significantly from near perfection to zero with increasing complexity when measured by either problem size (Figure 2(a))or

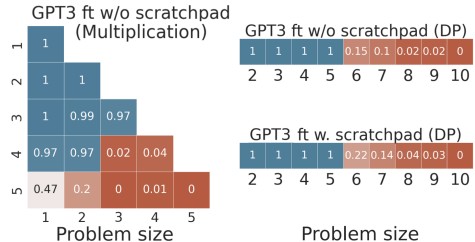

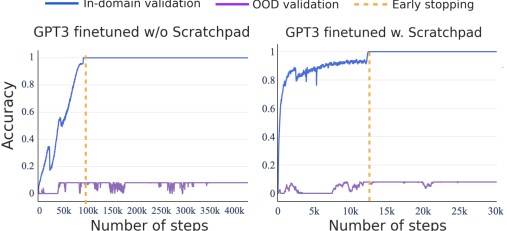

Figure 3: GPT3 finetuned exhaustively on task-specific data up to a certain problem size. The **blue** region represents the in-distribution examples and the **red** region refers to OOD examples. The same trend is observed for the puzzle task (See §B.2)

Figure 4: Results of training beyond the overfitting point for the multiplication task with the goal of exploring whether OOD generalization capabilities (i.e., *grokking*) arise.

average parallelism (Figure 2(b)).The trend remains the same for few-shot prompting (see §B.2). These results indicate that pre-training is in fact not sufficient to teach models how to combine basic operations to solve compositional problems, especially as problems grow more complex.

**Limits of transformers with question-answer training**    The limited performance of models may be attributed to the lack of task-specific data during pre-training. To fully bring out models' potentials in solving these tasks, we next exhaustively finetune GPT3 with question-answer pairs. In multiplication and DP, we finetune models with all enumerations of questions up to the maximum problem size[4] within reasonable training budget, leaving out 10% for validation and 10% for testing. In puzzles, we train on a subset of all instances up to $(K, M) \leq (4, 4)$ due to combinatorial explosion. We separately finetune GPT3 models on ∼1.8M multiplication pairs, ∼142K DP pairs, and ∼41K puzzle pairs (see details in §B.3). Additionally, to examine problems of different complexity, we consider different training splits based on the depth and width of computation graphs.

Figure 3 and Figure 5a show high accuracy for examples with splits seen during training, i.e., *in-domain*. However, the performance sharply declines when evaluating unseen splits during training, i.e., *out-of-domain (OOD)*. Similar trends hold in all tasks (see § B.3), suggesting that systematic problem-solving capabilities do not emerge via exhaustive training on task-specific data.

**Limits of transformers with explicit scratchpad training**    Next, we test whether we can explicitly teach models the required computational operations via *scratchpads*. To do so, we finetune GPT3 with question-scratchpad pairs for all tasks. We consider the same distribution splits as before. The results, presented in Figure 5b, show that once again GPT3 achieves near-perfect performance on in-distribution, but fails entirely in generalizing to OOD cases—in particular, wider or deeper computation graphs. These results indicate that even when training directly with guidance on the computation steps, models still fail to learn component operations in a generalizable manner. This observation holds for all tasks (See details in § B.4). Similarly, prompting transformers with question-scratchpad pairs enhances the performance compared to the zero-shot setting (refer to § B.5). However, this performance boost diminishes to zero as complexity increases. These findings suggest that the autoregressive characteristic of transformers, which forces them to tackle problems sequentially, presents a fundamental challenge that cannot be resolved by instructing the model to generate a step-by-step solution. Instead, models depend on a greedy process of producing the next word to make predictions without a rigorous global understanding of the task.

**Limits of transformers with grokking**    We explore whether extended training beyond overfitting leads to improved generalization abilities, a phenomenon known as grokking [59, 53]. Due to budget constraints, we only experiment on the multiplication task. Following [53], we fine-tune GPT3 with question-answer pairs for 420K steps and separately finetune GPT3 with question-scratchpad pairs for 30K steps. Both models' training far exceeded the point at which in-domain accuracy plateaus[5]. Figure 4 shows no improvement in generalization for OOD cases beyond the overfitting point, even

---

[4]We consider all $k_1$-by-$k_2$ digit multiplications with $1 \leq k_1, k_2 \leq 4$ and $k_1 \cdot k_2 \leq 9$; and all DP problems up to 5 elements. We selected sizes based on budget constraints for GPT3 finetuning, see §B.3 for cost details.

[5]The training duration for question-answer pairs is equivalent to 60 epochs and costs 50,000 USD. Training on question-scratchpad pairs was conducted for 40 epochs and costs 40,000 USD.

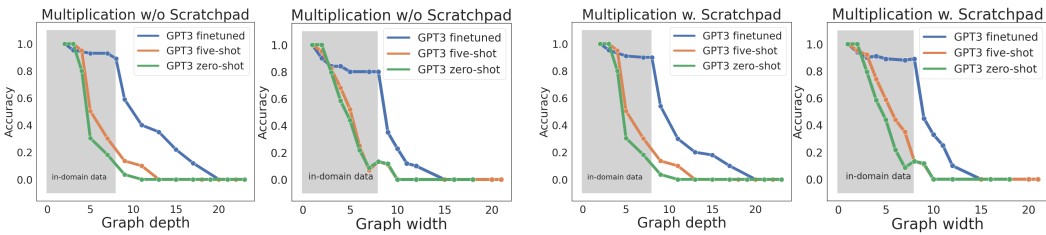

(a) Results on **question-answer** pairs.  (b) Results on **question-scratchpad** pairs.

Figure 5: **GPT3 finetuning and prompting accuracy** on different data splits. Although the in-distribution performance is almost perfect, GPT3 exhibits poor generalization with increasing graph depth and width. Refer to §B.3 and §B.4 for results on the puzzle and DP tasks.

after extensive training periods. We hypothesize that the absence of grokking may be due to the level of difficulty of the task. We speculate that increased task difficulty significantly impedes learning a well-structured representation, which, according to [47], aligns with achieving grokking. Even if grokking were to emerge through more prolonged training, such an approach would prove inefficient and unscalable. Future work is required to accurately explain when and how grokking occurs.

## 3.2 Breaking Down Successes and Failures of Transformers

### 3.2.1 Information Gain Explains Where Transformers Partially Excel

At times transformers predict partially correct answers even when the overall response is incorrect. We speculate that this may be due to particularities in the task distribution that allow for guessing partial answers without performing the full multi-step reasoning that the task requires.

Using relative information gain (defined in §2.3), we can predict surface patterns that a model is likely to learn and contrast them empirically. For multiplication, relative information gain shows that the first digit (two digits) of the output highly correlates with the first digit (two digits) of each input number (see §C.1). Hence, this spurious pattern is likely to be learned by a model. Similarly, the prediction of the last digit (or two digits) of the output is observed to solely rely on the last digit (or two digits) of each input number. This pattern holds true due to the principles of modulo arithmetic, which ensures the validity of this relationship in all cases. Empirically, we verify that models indeed learn the patterns we predicted and other patterns as well (e.g., order of magnitude of the answer, number of trailing zeros for multiplication) in all the settings with and without scratchpad. See details for multiplication, plus dynamic programming task analysis in §C.

These experiments suggest that if an output element heavily relies on a single or a small set of input features, transformers are likely to recognize such correlation during training and directly map these input features to predict the output element in testing, without going through the rigorous multi-hop reasoning and giving a false illusion of performing compositional reasoning.

### 3.2.2 Transformers Reduce Multi-Step Compositional Reasoning into Linearized Subgraph Matching

We now explore whether models' correct predictions on unseen test data are due to learning the underlying algorithm or, instead, explainable by exposure to similar training examples. We hypothesize that, beyond simple memorization, transformers largely rely on pattern matching for solving these tasks. To test this, we calculate the average frequency with which partial computations needed to solve an instance appear in the training data, for both correctly and wrongly predicted examples.

Given a model-generated computation graph $\widehat{G}_{A(\mathbf{x})}$ we analyze how often the full computation of each node $v \in \widehat{V}$ is seen in training. We define $v$'s *full computation* as the subgraph induced by all ancestors of $v$ including $v$, denoted $FC_{\widehat{G}_{A(\mathbf{x})}}(v)$. We say that $FC_{\widehat{G}_{A(\mathbf{x})}}(v)$ is seen during training if $FC_{\widehat{G}_{A(\mathbf{x})}}(v) \cong FC_{G_{A(\mathbf{x}')}}(w)$ for some computation graph $G_{A(\mathbf{x}')}$ in training, and for some $w \in V$. We characterize complexity of a full computation subgraph by its depth, as defined in §2.1.

Figure 6 shows that full computation subgraphs appear significantly more frequently in the training data for correctly predicted test examples than for incorrectly predicted ones, for both the multi-

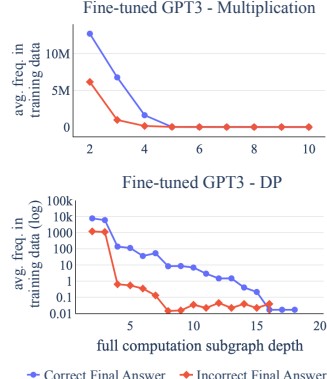

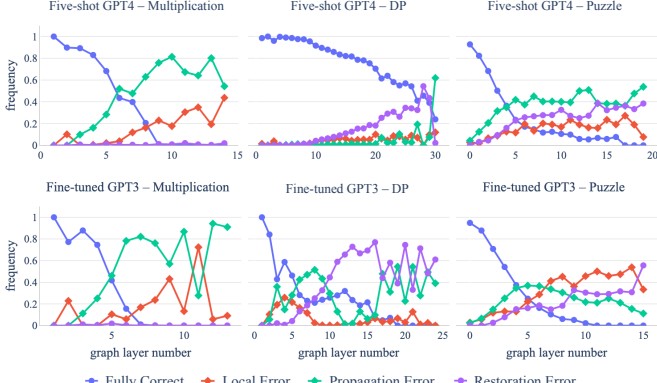

Figure 6: Average frequency in which test examples' full computations subgraph appear in the training data w.r.t. the subgraph depth, grouped by final answer.

Figure 7: Ratio of nodes in each of the four correct/error categories for each layer in computation graph. Results shown are for few-shot prompting and fine-tuning with scratchpad.

plication and DP task (both frequencies tend to zero for large depths since we ensured a disjoint train/test split). This high correlation suggests that pattern matching—and not general reasoning capabilities—may be the cause behind correct model outputs. This type of learning could be largely effective when the compositional complexity of tasks is low but it becomes less efficient when tasks are increasingly complex. This may elucidate the observed performance gain in low-complexity and in-domain cases and the striking performance drop in OOD and highly complex cases.

### 3.2.3 What Types of Errors do Transformers Make at Different Reasoning Depths?

For clearer understanding of where transformers fall short, we analyze the types of errors that transformers make for nodes at different layers in the computation graph. For every input $\mathbf{x}$, we compare the ground truth computation graph $G_{A(\mathbf{x})}$ with the (possibly incorrect) model-generated computation graph $\widehat{G}_{A(\mathbf{x})}$. We consider a node $v$ as having a *correct value* if and only if $s(v) = \widehat{s}(v)$.[6]. We consider a node $v$ to be derived from a *correct computation* if given that $U = \{u_1, \ldots, u_k\}$ are the immediate predecessors of $v$ in $\widehat{G}_{A(\mathbf{x})}$ and that $\widehat{op}(v) = f$, we have that $f(u_1, \ldots, u_k) = \widehat{s}(v)$. Note that the notion of correct computation is independent of $G$, and that a node $v$ derived from a correct computation may not have the correct value if an error occurred in some of its ancestors.

We classify each node $v \in \widehat{V}$ into one of four categories. Node $v$ is **fully correct** if $v$ and its ancestors have correct values and are derived from correct computations. If a node $v$ is not fully correct, its error can be of the following types: $v$ has a **local error** if its parent nodes have correct values but $v$ is derived from an incorrect computation (i.e., a one-hop reasoning error); $v$ has a **propagation error** if $v$ is derived from a correct computation but some of its parent nodes have incorrect values; $v$ has a **restoration error** if it has a correct value but is derived from an incorrect computation.

Figure 7 shows results for few-shot GPT4 and fine-tuned GPT3 with scratchpad, with respect to graph layer number for each node. In all settings, the ratio of fully correct nodes is almost perfect but sharply decreases toward zero with increasing graph layers. Moreover, the ratio of propagation errors is usually higher than the ratio of local errors. Both phenomena suggest that models are able to correctly perform single-step reasoning, potentially due to memorizing such single-step operations during training, but fail to plan and compose several of these steps for an overall correct reasoning.

Both the DP and the puzzle tasks have a high ratio of restoration errors, suggesting memorization since correct outputs are produced despite incorrect computations. There are signs of memorization even when restoration errors are near zero: 82.3% of the final correct answers for 4-digit by 2-digit multiplications (a setting unseen during training) had at least one error in the computation graph, but still produced correct answers. These patterns are possibly due to high frequency of (input, output) multiplication pairs in the pretraining data, in contrast to intermediate reasoning steps.

---

[6]If a node $v$ does not appear in the ground truth graph $G$, we consider it to have an incorrect value.

# 4 Error Propagations: The Theoretical Limits

Experiments (§3) highlight the limitations of current transformers in handling complex, multi-step reasoning tasks. Concretely, we show that errors rapidly escalate as the problem size grows (§3.2.3). Here, we aim to provide theoretical insights into why autoregressive transformer LLMs can perform significantly worse in compositional tasks as the problem size increases, making explicit the different ways in which compounding stochastic errors affect final performance. We argue using stylized examples that transformers may be too limited to solve compositionally complex tasks. Formal statements and full proofs are provided in §D.

Algorithms designed to solve compositional tasks typically involve multiple independent applications of a function and/or iterated applications of the same function. A transformer executing such an algorithm acts as an estimator of these functions. In this context, we examine the probability of such an estimator reaching the correct answer as the problem size increases. We first consider a scenario where a transformer estimates an algorithm requiring $n$ independent applications of a function:

**Proposition 4.1** (informal). *Let $f_n$ involve the combination $h_n$ of $n$ independent applications of a function $g$. Let $\widehat{f}, \widehat{g}, \widehat{h}_n$ be their estimators. Assume that $\widehat{h}_n$ is a perfect estimator of $h_n$ and that $h_n$ has low collision, with $c_n$ being an upper bound of $h_n$'s collision rate ($c_n < c \; \forall n$, with $c \ll 1$). If $\mathbb{P}(g \neq \widehat{g}) = \epsilon > 0$ where $\widehat{g}$'s errors are independent, then $\mathbb{P}(f_n \neq \widehat{f}_n) > 1 - c_n - (1 - \epsilon)^n \cdot (1 - c_n)$. This implies that $\mathbb{P}(f_n \neq \widehat{f}_n)$ decreases **exponentially** as $n$ increases, with $\liminf_{n \to +\infty} \mathbb{P}(f_n \neq \widehat{f}_n) \geq 1 - c$. Moreover, if $c_n \leq \beta \alpha^n$ for some $\alpha \in (0, 1), \beta > 0$, $\mathbb{P}(f_n \neq \widehat{f}_n)$ tends exponentially to 1 as $n$ increases.*

Prop. 4.1's proof (§D.1) shows the rate of convergence is exponential, thus concluding that transformers will rapidly fail with increasing $n$. Let's now analyze the iterated application function scenario.

**Proposition 4.2** (informal). *Let $f_n(\mathbf{x}) = g^n(\mathbf{x})$ involve the repeated application of $g$. Assume that the probability of recovering from a mistake due to the randomness of applying the estimator on an incorrect input has probability at most $c$. If $\mathbb{P}(g \neq \widehat{g}) = \epsilon > 0$, then $\mathbb{P}(f_n \neq \widehat{f}_n)$ decreases **exponentially** with $n$. Precisely, $\mathbb{P}(f_n \neq \widehat{f}_n) \geq 1 - (1 - \epsilon - c)^{n-1}(1 - \epsilon - c/(c + \epsilon))$, implying $\liminf_{n \to +\infty} \mathbb{P}(f_n \neq \widehat{f}_n) \geq 1 - c/(c + \epsilon)$.*

The argument is as follows. Let $s_n := \mathbb{P}(f_n = \widehat{f}_n)$, where $s_1 = 1 - \epsilon$ by definition. Derive $s_n \leq (1 - \epsilon - c) \cdot s_{n-1} + c$ using law of total probability. Then, prove by induction a non-recursive upper bound for $s_n$ with limit $\frac{c}{c+\epsilon}$ when $n \to +\infty$. See formal statement and derivation in §D.2.

Prop. 4.2's proof also shows an exponential rate of convergence. Note that if $c \ll \epsilon$ then $\liminf_{n \to +\infty} \mathbb{P}(f_n \neq \widehat{f}_n) \approx 1$. It is reasonable to assume $c \ll \epsilon$ when $g$ has low collision, since $c$ represents the probability of the estimator $\widehat{g}(y)$ arriving at the correct output $g(x)$ by chance when given the wrong input $y \neq x$. More details in §D.3.

Moreover, repeated applications of a function often imply unbounded errors: if $g(x)$ can be expressed as an affine transformation $Fx + c$, then it may be viewed as a first-order vector autoregression, which are known to be unstable when $|\lambda| \geq 1$ for at least one $\lambda$ eigenvalue of $F$ [31, Prop. 10.1]. While we make these arguments with affine maps, similar behaviors, possibly even more acute, could occur with nonlinear maps [25]—but their study is beyond the scope of this paper.

In Prop. 4.2's current form, we implicitly assume that there is a single valid reasoning for each input since $g$ is a function. We can potentially generalize this assumption with a state-transition framing, where the probability of transitioning from a valid state to an invalid one is $\epsilon$, and the probability of recovering from an invalid state is at most $c$. See formal statement in D.2.

All tasks evaluated in the present work can be seen as instances of the results just proven. Prop. 4.1 directly applies to multiplication, since $m$-by-$n$ digit multiplication can be seen as $n$ independent instances of $m$-by-1 digit multiplication (see Cor. D.1). Prop. 4.2 directly applies to the recursive function of the dynamic programming task, as well as to $m$-by-1 digit multiplication, and to the puzzle through its elimination function (details in D.3). They are also all low collision settings.

Note that Prop 4.1 and 4.2 apply to any high-performant estimator of reasoning tasks. We focus on out-of-the-box transformers to align with the scope of our experiments and with the goal of framing empirical results. In §5, we discuss how these propositions may inform future research directions.

# 5 Discussion

**Collapsed Compositionality and Robustness Implications** Transformers today demonstrate undeniably powerful empirical results. Yet, our study suggests that they may have fundamental weaknesses in certain intellectual tasks that require true multi-step compositional operations such as multiplications and logic puzzles. Our careful study based on the computation graph and analyses demonstrates that transformers can often solve multi-step compositional problems by collapsing the depth of the compositional operations via analogical pattern matching. More broadly, our findings suggest that the strong performance of transformers should be taken with a certain grain of salt: Despite initially appearing challenging, certain tasks may not possess the inherent compositionality they seem to have. This is due to the fact that desired solutions could be readily derived from input-output sequences present in the training data, allowing for shortcut pattern matching to produce acceptable solutions. However, such an approach can ultimately result in poor generalization as shown in our study. For example, fine-tuning GPT3 on our tasks both with and without explicit reasoning graphs shows that models' learning fails to generalize beyond levels of complexity seen in training.

**Theoretical Findings and their Empirical Implications** The proofs presented in §4 show that, under reasonable assumptions, the probability of incorrect predictions converges exponentially to $\approx 1$ for abstract compositional tasks. Importantly, these proofs apply to autoregressive LMs in general. Our insights indicate that the current configuration of transformers, with their reliance on a greedy process for predicting the next word, constrains their error recovery capability and impedes the development of a comprehensive global understanding of the task. Building on these findings, we suggest several empirical strategies for harnessing the potential of transformers. Firstly, transformers may be employed in ways that require chaining only a few compositional steps to reach a solution rather than lengthy reasoning steps (e.g., [35]). Secondly, transformers may be best suited for compositional tasks where evaluation metrics can afford some leniency; for example, finding approximate solutions that do not require executing the whole graph, such as identifying the most significant digit in a multiplication. Finally, we suggest augmenting transformers with planning modules as well as using refinement methods, that can iteratively improve their generations [82, 48].

**Call for Broad Participation to Investigate Limitations** Identification of limitations is an important step towards achieving greater robustness. Our study suggests fundamental limitations that impede transformers from fully mastering certain compositional operations. However, we acknowledge that due to our compute budget constraints as well as limited access to the largest language models such as GPT4, we are unable to push the empirical limits of transformers even further in terms of training data size and number of epochs. We invite the broader research community, particularly those with more extensive resources at their disposal, to investigate these possibilities further.

# 6 Related Work

**Reasoning abilities in transformer LLMs** Recently, transformers [11, 58, 57, 17, 16, 63, 73, 74] have demonstrated impressive reasoning abilities across a wide range of tasks, even outperforming humans in certain cases [79, 28, 15, 85]. This success has been largely attributed to the scaling effect, where larger models and training datasets result in improved performance [38, 33, 1]. However, these models have also been shown to struggle across multiple domains [32], including algorithmic reasoning [78], commonsense reasoning [62, 40], theory of mind [65], planning [76], logical reasoning [70], and ethical reasoning [36]. These difficulties have motivated us to take a step back and thoroughly examine both the successes and failures of transformers from empirical and theoretical perspectives on compositional reasoning tasks.

**Challenges of transformers in compositional tasks** Transformers perform fairly well in single-step reasoning tasks [70], but face challenges when it comes to effectively combining multiple steps to solve compositionally complex problems [84, 55, 66, 81]. Recent research has focused on overcoming these limitations through various approaches. First, fine-tuning transformers to directly generate the final answer while keeping the reasoning implicit [7, 18]. Second, encouraging transformers to generate reasoning steps explicitly within a single generation [55, 80, 44, 42]. For example, Nye et al. [55] and Zhou et al. [86] used scratchpads to teach transformers how to perform algorithmic reasoning tasks such as addition by splitting the task into intermediate steps [44, 80]. Further, leveraging LLMs to generate each reasoning step iteratively via a selection and inference mechanism [20, 19, 72].

Lastly, choosing a training split that maximizes the number of observed patterns between the train and test data [10], or diversifying in-prompt examples to cover the maximum of patterns [41], ultimately enhancing generalization. The primary focus of these studies is to enhance model performance on compositional problems without striving for complete mastery. In contrast, our work explores the fundamental limits of vanilla transformers in achieving full mastery, striving for 100% performance in both in-domain and OOD settings. Our findings show that reaching full mastery is inherently challenging, providing insights into the complexities involved.

**Challenges of transformers in generalization**  Extensive research has been done to investigate the generalization capabilities of transformers [3, 54, 26, 64, 8, 46]. This encompasses various facets of generalization, including easy-to-hard generalization [67, 4], length generalization [2, 60, 13, 54, 8], and generalization on symbolic mathematical integration [83]. Schwarzschild et al. [67] and Bansal et al. [4] employ weight-tied neural networks to generalize from easy to hard examples. Liu et al., [45] found that shallow transformers learn shortcuts during training, leading to poor OOD generalization. Razeghi et al. [64] revealed a positive correlation between the frequency of training terms and their test performance. Building upon this line of inquiry, we present a more rigorous examination of sub-graph matching between training and test instances for complex compositional tasks where we demonstrate how pattern matching can hinder generalization. We complement our empirical results with theoretical insights on transformers' limits.

**Grokking**  The phenomena of models' gaining generalization capabilities when training significantly beyond overfitting, known as *grokking* was recently introduced in [59]. Subsequent works focus on characterizing when and why grokking arises: [47] show that perfect generalization in an arithmetic addition task happens when there is sufficient data to determine the appropriate structured representation, later extended to sparse parity in [52] where a sparse subnetwork of neurons is shown responsible for generalization behavior. Recently, [77] propose that grokking occurs when a task admits a generalizing and a memorizing solution, and the former is slower to learn. In this present work, our aim is not to explain grokking but rather to observe its emergence. We do not observe grokking arising in the context of multiplication, and we leave it to future work to explore whether this may be due to task difficulty hindering the learning of well-structured representations.

**Transformers' theoretical expressiveness**  Lin et al. [43] study autoregressive models' limitations from a computational complexity theory perspective. Transformer-specific work has focused on quantifying the class of problems that (not necessarily autoregressive) transformers can express assuming perfect parameters [51, 50, 14, 49, inter alia]. All tasks analyzed in our work belong to a class expressible by transformers, suggesting that known upper bound might not be tight. Importantly, Hahn [30] shows that transformers cannot robustly model noncounter-free regular languages even when allowing infinite precision. In contrast, our focus is on error accumulation, which enables to investigate if reasoning tasks theoretically solvable by transformers are likely to be solved by them.

Additional literature and societal impact discussion can be found in §E.

# 7  Conclusions

On a broader scope, as transformers continue to gain widespread deployment with significant real-world impacts, it is ever more urgent to understand their successes and failures. Our study critically investigates transformers' limitations and emphasizes the need to develop models capable of robust generalization and systematic problem-solving. By examining the compositional capabilities of these models, we aspire to work towards more reliable AI systems that excel not only in tasks where abundant training examples are sufficient, but also in cases requiring precise compositional reasoning.

# 8  Limitations

We focus on analyzing compositional reasoning capabilities through the lens of computation graphs. Although they are a useful way to systematically represent rigorous reasoning processes, it is important to note that for the scratchpad approach, we are limited to only establishing a correlation between the model generation and its preceding context, as we cannot inspect the exact tokens model attends to when making the prediction. This limitation arises from our lack of access to the activations of the studied models. Furthermore, we posit that alternative approaches to linearizing reasoning processes may yield different performances and provide opportunities for further exploration.

## Acknowledgements

We thank members of the Mosaic team at AI2 for valuable feedback on this project, as well as Agustín Santiago Gutiérrez and Kawin Ethayarajh for valuable discussions. This research was supported by the NSF DMS-2134012, DARPA MCS program through NIWC Pacific (N66001-19-2-4031), and the Allen Institute for AI.

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

# Appendices

# A    Compositional Tasks

## A.1    Multiplication

**Data Construction**    We exhaustively generate multiplication problems as question-answer pairs (e.g., Q: "What is 4 times 32?" A: "128"). We focus on multiplications of two numbers $x = (x_1, x_2, \ldots, x_k)$ and $y = (y_1, y_2, \ldots, y_k)$ where each number can have up to $k$ digits, amounting to $9 \times 10^{(k-1)}$ combinations per each number. We set $k$ to 5 in our experiments. Figure 8 showcases an example prompt for performing few-shot learning without the inclusion of a scratchpad, while Figure 9 demonstrates an example prompt using a scratchpad. Throughout our experimentation, we explored various versions of the scratchpad, ranging from verbose and detailed to more concise alternatives. Among these variations, the scratchpad version depicted in Figure 9 ultimately produced the most favorable outcomes. Listing 1 shows the Python code for solving the task.

```
To multiply two numbers, start by multiplying the rightmost digit of the
multiplicand by each digit of the multiplier, writing down the products and
carrying over any remainders.  Repeat this process for each digit of the
multiplicand, and then add up all the partial products to obtain the final
result.

Questions: what's 22 times 2? Answer 44.
```

Figure 8: Example prompt for the multiplication task used for the few-shot setting.

```
Question: What is 35 times 90?

Scratchpad: Let's perform the multiplication step by step:

Let's multiply 35 by the digit in the ones place of 90, which is 0.

1. Multiply 0 by the digit in the ones place of 35, which is 5. This gives 5 x 0
= 0. Write down the result 0.
2. Multiply 0 by the digit in the tens place of 35, which is 3. This gives 3 x 0
= 0. Write down the result 0.
3. The partial product for this step is A=0 which is the concatenation of the
digits we found in each step.

Now, let's multiply 35 by the digit in the tens place of 90, which is 9.

4. Multiply 9 by the digit in the ones place of 35, which is 5. This gives 5 x 9
= 45. Write down the result 5 and carry over the 4 to the next step.
5. Multiply 9 by the digit in the tens place of 35, which is 3. Add the carryover
from the previous step to account for this. This gives (3 x 9) + 4 = 31. Write
down the result 31.
6. The partial product for this step is B=315 which is the concatenation of the
digits we found in each step.

Now, let's sum the 2 partial products A and B, and take into account the position
of each digit: A=0 (from multiplication by 0) and B=315 (from multiplication by 9
but shifted one place to the left, so it becomes 3150). The final answer is 0 x 1
+ 315 x 10 = 0 + 3150 = 3150.
```

Figure 9: A sample scratchpad for the multiplication task.

```
1  def multiply(x, y):
2      summands = [0] * len(y)
3      for i in range(len(y) - 1, -1, -1):
```

```
4        digits = [0] * len(x)
5        carry = 0
6        for j in range(len(x) - 1, -1, -1):
7            t = x[j] * y[i]
8            t += carry
9            carry = t // 10
10           digits[j] = t % 10
11       digits.insert(0, carry)
12       summands[i] = sum(digits[-k] * (10 ** (k - 1)) for k in range
     (1, len(digits) + 1))
13
14   product = sum(summands[-i] * (10 ** (i - 1)) for i in range(1, len
     (y) + 1))
15   return product
```

Listing 1: Example Python code for solving the multiplication task.

## A.2 Einstein's Puzzle

**Data Construction** In our experiments, we initially establish a set of properties, such as Color, PhoneModel, Pet, and so forth, along with their corresponding values expressed in natural language templates (e.g., "The house has a red color."). We then devise a fundamental and straightforward set of clue types: 1) 'found_at', e.g., "Alice lives in House 2", 2) 'same_house', e.g., "The person who is a cat lover lives in the house that has a red color.", 3) 'direct_left', e.g., "The person who has a dog as a pet lives to the left of the person who lives in a red house.", and 4) 'besides', e.g., "The person who has a dog as a pet and the person who has a red house live next to each other." In addition, we also set up harder clue types such as 'not_at', 'left_of' (not necessarily directly left of), 'two_house_between', etc. which are only used in auxiliary experiments.

The solution to the puzzle is a matrix of size $K \times M$, where $K$ represents the number of houses and $M$ the number of attributes. During the puzzle generation, the $M$ properties are randomly selected from the candidate pool, followed by the random sampling of $K$ values for each property. The sampled values are then randomly permuted and assigned within the table to create the solution. It is important to note that we ensure one of the sampled properties is 'Name' to enhance the readability and comprehensibility of the puzzles. To construct the clues, we initially over-generate all valid clues based on the solution and subsequently remove redundant clues at random until we obtain a set with a

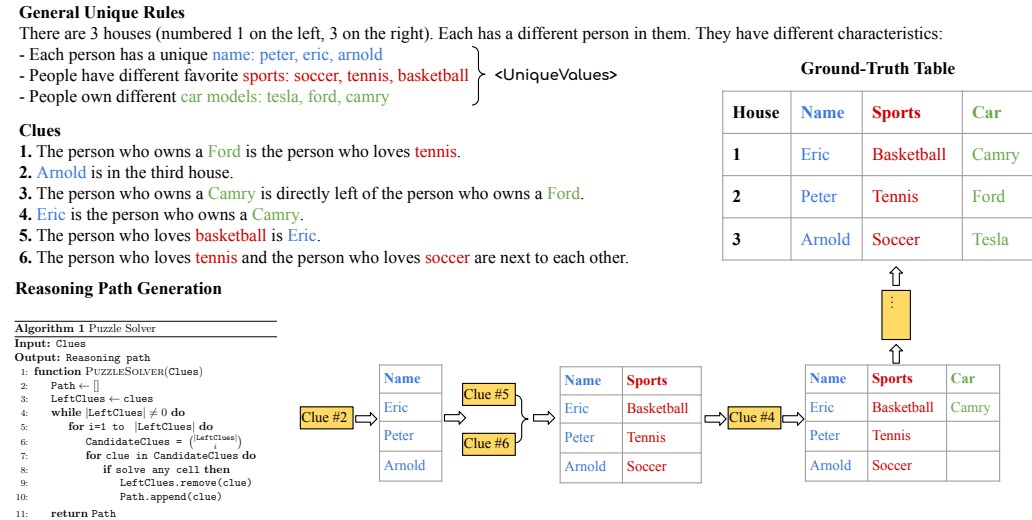

Figure 10: A sample of the puzzle task and the reasoning path to reach a solution.

```
This is a logic puzzle. There are 3 houses (numbered 1 on the left, 3 on the
right). Each has a different person in them. They have different characteristics:
- Each person has a unique name: peter, eric, arnold
- People have different favorite sports: soccer, tennis, basketball
- People own different car models: tesla model 3, ford f150, toyota camry

1. The person who owns a Ford F-150 is the person who loves tennis.
2. Arnold is in the third house.
3. The person who owns a Toyota Camry is directly left of the person who owns a
Ford F-150.
4. Eric is the person who owns a Toyota Camry.
5. The person who loves basketball is Eric.
6. The person who loves tennis and the person who loves soccer are next to each
other.

Let's think step by step. Please first briefly talk about your reasoning and show
your final solution by filling the blanks in the below table.

$ House: ___ $ Name: ___ $ Sports: ___ $ Car: ___
$ House: ___ $ Name: ___ $ Sports: ___ $ Car: ___
$ House: ___ $ Name: ___ $ Sports: ___ $ Car: ___

Reasoning:
Step 1: First apply clue <Arnold is in the third house.> We know that The Name in
house 3 is arnold.
Step 2: Then combine clues: <The person who loves tennis and the person who loves
soccer are next to each other.> <The person who loves basketball is Eric.>
Unique Values Rules and the fixed table structure. We know that The Name in house
1 is eric. The FavoriteSport in house 1 is basketball. The Name in house 2 is
peter.
Step 3: Then apply clue <Eric is the person who owns a Toyota Camry.> We know
that The CarModel in house 1 is toyota camry.
Step 4: Then apply clue <The person who owns a Toyota Camry is directly left of
the person who owns a Ford F-150.> and Unique Values We know that The CarModel in
house 2 is ford f150. The CarModel in house 3 is tesla model 3.
Step 5: Then apply clue <The person who owns a Ford F-150 is the person who loves
tennis.> and Unique Values We know that The FavoriteSport in house 2 is tennis.
The FavoriteSport in house 3 is soccer.
The puzzle is solved.

Final solution:
$ House: 1 $ Name: Eric   $ Sports: Basketball $ Car: Camry
$ House: 2 $ Name: Peter  $ Sports: Tennis     $ Car: Ford
$ House: 3 $ Name: Arnold $ Sports: Soccer     $ Car: Tesla
```

Figure 11: A sample scratchpad for the puzzle task.

unique solution, as previously sampled. This process ensures a coherent and engaging puzzle-solving experience. Refer to Figure 10 for an example.

**Graph Construction Algorithm**   To solve the complex compositional reasoning process for a logical grid puzzle, we use existing puzzle solvers [21] to generate the computation graph. It follows the basic greedy principle of applying the minimum number of rules to solve any cell, i.e., if using only one rule to solve any given cell, then apply this rule. This algorithm iterates through all clues in the clue set until one or a set of clue combinations can solve any cell in the table. While it may not be the most efficient way to solve the puzzle, it provides models with explicit scratchpad verbalization through an intuitive computation graph. Refer to Figure 10 for the pseudo-code of the process, and Figure 11 for a scratchpad example.

## A.3 Dynamic Programming Problem

### A.3.1 Solution to this problem

Let $a = [a_1, \ldots, a_n]$ be an input. Let $dp_i$ be the maximum sum of a subsequence that does not include adjacent elements, when considering only the elements of the input from the $i$-th position onwards.

Trivially, $dp_n = \max(a_n, 0)$ since we only want to choose a number if it is non-negative. Moreover, $dp_{n-1} = \max(a_n, a_{n-1}, 0)$ since we cannot choose adjacent numbers.

For any given $dp_i$ with $i \leq n - 2$, we can express it in terms of $dp_{i+1}$ and $dp_{i+2}$. Concretely, the maximum sum of a subsequence starting at position $i$ may or may not include the element in the $i$-th position, $a_i$. If the subsequence includes $a_i$, then the maximum sum is $a_i + dp_{i+2}$, since using $a_i$ blocks us from using the next element. If the subsequence does not include $a_i$, then its sum is $dp_{i+1}$. Moreover, the answer may never be less than zero, because otherwise we would select the empty sequence[7]. In summary,

$$dp_i = \max(dp_{i+1},\ a_i + dp_{i+2},\ 0)$$

We now have a recursion with its base cases $dp_n = \max(a_n, 0)$ and $dp_{n-1} = \max(a_n, a_{n-1}, 0)$, and we can therefore compute all values in $O(n)$. It now only rests to reconstruct the lexicographically smallest subsequence that maximizes the desired sum, based solely on the computed $dp$ values.

Starting from $dp_1$ and iterating sequentially through $dp_{n-2}$, we choose an item if and only if $dp_i = a_i + dp_{i+2}$ (that is, the maximum sum comes from choosing the current element) and we have not chosen the previous element. This helps disambiguate cases where choosing or not choosing $a_i$ yields the same sum, but possibly only one of those will not incur in choosing adjacent numbers. Similarly, for positions $i = n - 1$ and $i = n$ we choose the element if $dp_i = a_i$ (that is, choosing the element yields the maximum sum) and we have not chosen the immediately previous element. See an example Python solution in 2.

> Given a sequence of integers, find a subsequence with the highest sum, such that no two numbers in the subsequence are adjacent in the original sequence.
>
> Output a list with "1" for chosen numbers and "2" for unchosen ones. If multiple solutions exist, select the lexicographically smallest. input = [3, 2, 1, 5, 2].

Figure 12: Example prompt for the DP task, used for zero-shot and few-shot settings.

```python
def maximum_sum_nonadjacent_subsequence(arr):

    N = len(arr)
    dp = [0 for _ in range(N)]

    dp[N - 1] = max(arr[N - 1], 0)
    dp[N - 2] = max(max(arr[N - 1], arr[N - 2]), 0)

    for i in range(N - 3, -1, -1):
        dp[i] = max(max(dp[i + 1], arr[i] + dp[i + 2]), 0)

    # reconstruct the answer with a fixed-size graph
    result = []
    can_use_next_item = True

    for i in range(N - 2):
        if dp[i] == arr[i] + dp[i + 2] and can_use_next_item:
            result.append(1)
            can_use_next_item = False
        else:
```

---

[7]We don't need to explicitly check for this since $dp_n \geq 0$. However, we include the condition to ease the scratchpad logic.

```
21            result.append(2)
22            can_use_next_item = True
23
24      if dp[N - 2] == arr[N - 2] and can_use_next_item:
25          result.append(1)
26          can_use_next_item = False
27      else:
28          result.append(2)
29          can_use_next_item = True
30
31      if dp[N - 1] == arr[N - 1] and can_use_next_item:
32          result.append(1)
33      else:
34          result.append(2)
35
36      return result
```

Listing 2: Example Python code for solving the DP task. We chose this implementation because the computation graph has always the same topology for any given input length.

```
Question: Let's solve input = [3, 2, 1, 5, 2].

Scratchpad: dp[4] = max(input[4], 0) = max(2, 0) = 2
dp[3] = max(input[3], input[4], 0) = max(5, 2, 0) = 5
dp[2] = max(dp[3], input[2] + dp[4], 0) = max(5, 1 + 2, 0) = 5
dp[1] = max(dp[2], input[1] + dp[3], 0) = max(5, 2 + 5, 0) = 7
dp[0] = max(dp[1], input[0] + dp[2], 0) = max(7, 3 + 5, 0) = 8

Finally, we reconstruct the lexicographically smallest subsequence that fulfills
the task objective by selecting numbers as follows. We store the result on a list
named "output".

Let can_use_next_item = True.
Since dp[0] == input[0] + dp[2] (8 == 3 + 5) and can_use_next_item == True, we
store output[0] = 1. We update can_use_next_item = False.
Since dp[1] != input[1] + dp[3] (7 != 2 + 5) or can_use_next_item == False, we
store output[1] = 2. We update can_use_next_item = True.
Since dp[2] != input[2] + dp[4] (5 != 1 + 2) or can_use_next_item == False, we
store output[2] = 2. We update can_use_next_item = True.
Since dp[3] == input[3] (5 == 5) and can_use_next_item == True, we store
output[3] = 1. We update can_use_next_item = False.
Since dp[4] != input[4] (2 != 2) or can_use_next_item == False, we store
output[4] = 2.

Reconstructing all together, output=[1, 2, 2, 1, 2].
```

Figure 13: A sample scratchpad for the DP task used for fine-tuning with few-shot settings.

**Data Construction**    We exhaustively generate data for this DP task. For question-answer setting, we include a thorough explanation of the task before asking to generate a solution (see Figure 12). We use all lists up to 5 elements as training, and we consider only lists where elements are in the range $[-5, 5]$ (giving a total of $11^n$ lists for an input list of size $n$). For out-of-domain evaluation, we use lists of sizes 6 to 10 inclusive. Example scratchpads and zero-shot prompts are shown in Figure 13 and 12 respectively. The scratchpad is generated automatically through templates. We considered five exemplars for the few-shot setup.

## B  Experimental Setups & Empirical Results

### B.1  Models

For our experiments, we evaluate the performance of 6 LLMs: GPT4 (`gpt-4`) [58], ChatGPT (GPT3.5-turbo) [57], GPT3 (`text-davinci-003`) [11], FlanT5 [17] and LLaMa [75]. The evaluations were conducted from January 2023 to May 2023 using the OpenAI API. We perform fine-tuning on GPT3 (`text-davinci-003`) for the three tasks, observing faster convergence when training on question-scratchpad pairs rather than question-answer pairs. For question-answer pairs fine-tuning, we train separately the model for {14, 12, 4} epochs for multiplication, puzzle, and DP respectively, saving the best model based on the validation set. Regarding training on question-scratchpad pairs, we train the model for {16, 8, 2} epochs for multiplication, puzzle, and DP. The batch size is set to approximately 0.2% of the number of examples in the training set. Generally, we observe that larger batch sizes tend to yield better results for larger datasets. For the learning rate multiplier, we experiment with values ranging from 0.02 to 0.2 to determine the optimal setting for achieving the best results and chose 0.2. During inference, we set nucleus sampling $p$ to 0.7 and temperature to 1. For each task, we evaluate the performance of each model on 500 test examples.

### B.2  Limits of Transformers in Zero- and Few-shot Settings

Figure 15, Figure 17 and Figure 20 show the zero-shot performance of GPT4, ChatGPT, LLaMA and FlanT5 on the three tasks. Overall, there is a notable decline in performance as the task complexity increases (measured by graph parallelism for multiplication and DP, and propagation steps for puzzles as shown in Figure14). The few-shot performance with question-answer pairs results in minimal improvement over the zero-shot setting as depicted in Figure 16 and Figure 20 for the multiplication and DP tasks. In contrast, the few-shot setting did not lead to any improvement in the puzzle task.

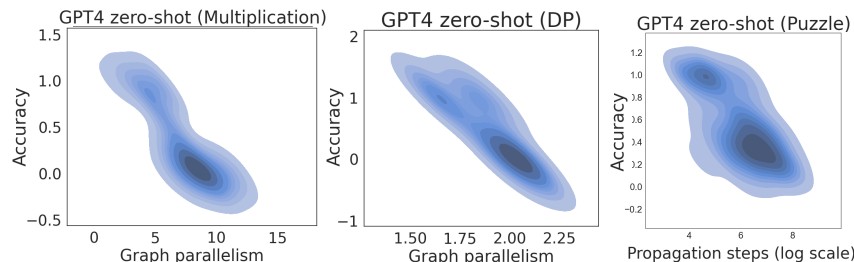

Figure 14: Graph parallelism vs accuracy. The accuracy decreases as the complexity increases.

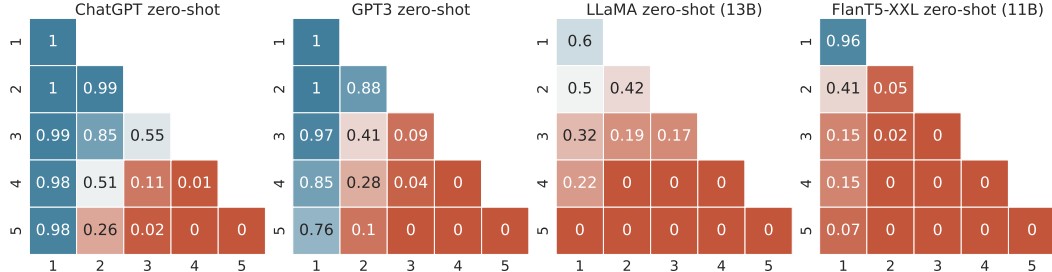

Figure 15: **Zero-shot accuracy**. Performance of ChatGPT, GPT3, LLaMA and FlanT5 on the **multiplication** task.

### B.3  Limits of Transformers with question-answer Training

Figure 18 and Figure 21 show the performance of GPT3 finetuned on question-answer pairs. The model was trained on various splits, considering the problem size, depth, and width of the computation graph. Specifically, for the multiplication task, the model was fine-tuned on a range of multiplication problems, spanning from 1-digit by 1-digit multiplication to 4-digit by 2-digit multiplication amounting to 1.8M pairs. As for the puzzle task, the model was fine-tuned on puzzles of sizes ranging from 2x2 to 4x4 resulting in a total of 142k pairs. Additionally, for the DP task, the model was fine-tuned on problems with a sequence length of 5 resulting in 41K pairs. In an additional setup, we divided

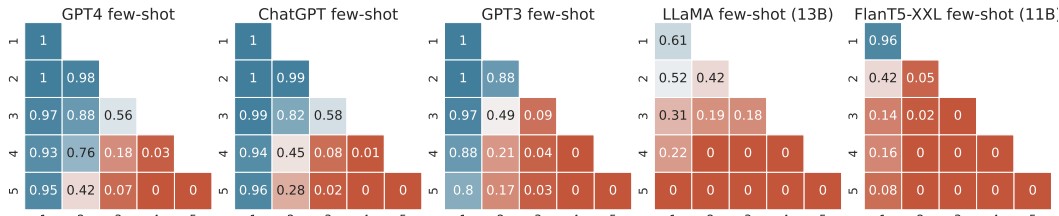

Figure 16: **Few-shot accuracy** with **question-answer** pairs. Performance of GPT4, ChatGPT, GPT3, LLaMA and FlanT5 on the **multiplication** task.

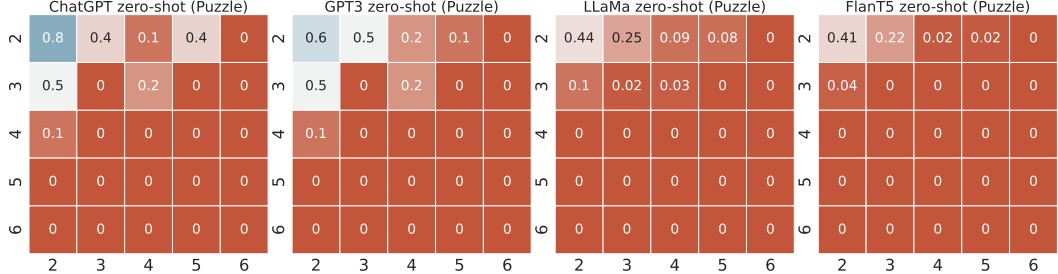

Figure 17: **Zero-shot accuracy**. Performance of ChatGPT, GPT3, LLaMA and FlanT5 on the **puzzle** task. Few-shot performance led to worse performance.

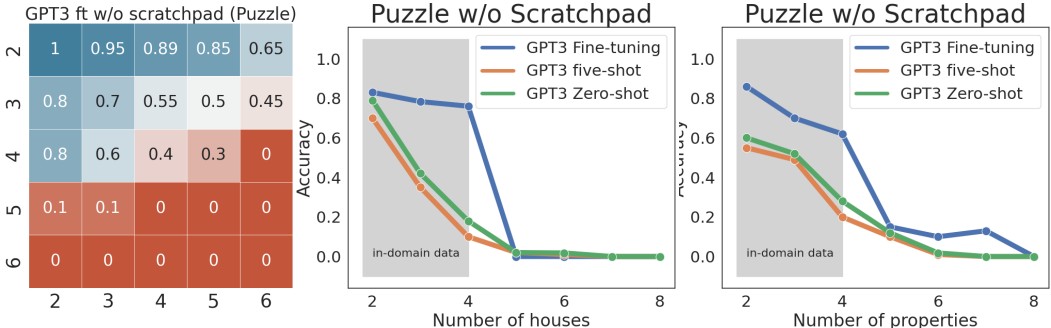

Figure 18: GPT3 finetuned on the puzzle task using **question-answer** pairs. The training data consisted of puzzles of size 4x4, and the model was subsequently evaluated on larger puzzle sizes for OOD testing.

those datasets based on the depth and width of the computation graph for all the tasks and finetuned on different splits. The results indicate a lack of generalization for out-of-domain (OOD) examples while showcasing near-perfect performance for in-domain examples. One hypothesis on why the model exhibit such a poor generaliztion is tokenization. So we train GPT2-XL from scratch on up to 4x4 (90M data points), we assign each digit to one token and each math symbol as well. However, the performance is still low and GPT2-XL fails to answer correctly 3x3 test examples.

**GPT3 finetuning cost** We will discuss here the approximate cost of fine-tuning GPT3 for the multiplication task. When fine-tuning with question-answer pairs, each example typically consists of around 20 tokens, and 250 tokens for question-scratchpad pairs. The cost for utilizing the `text-davinci-003` model amounts to $0.02 (USD) per 1,000 tokens. With this particular setup, the total number of training examples required for multiplication up to 5 digits by 5 digits reaches an astonishing figure of approximately 9.1 billion examples. Should we choose to fine-tune GPT3 for 4 epochs on question-answer pairs, the cost would amount to $12 million and $700 million for question-scratchpad training. For a more comprehensive breakdown of the cost per problem size, please refer to Table 1.

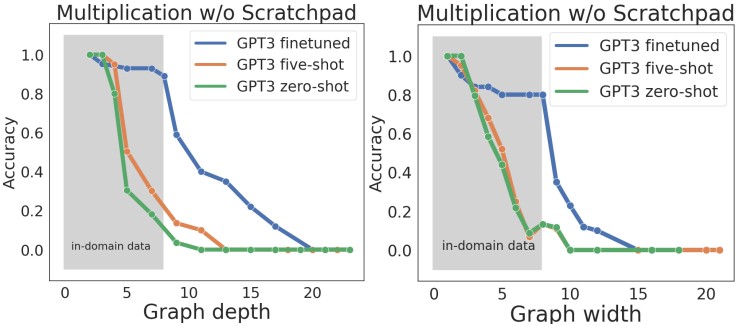

Figure 19: GPT3 finetuned on the multiplication task using **question-answer** pairs

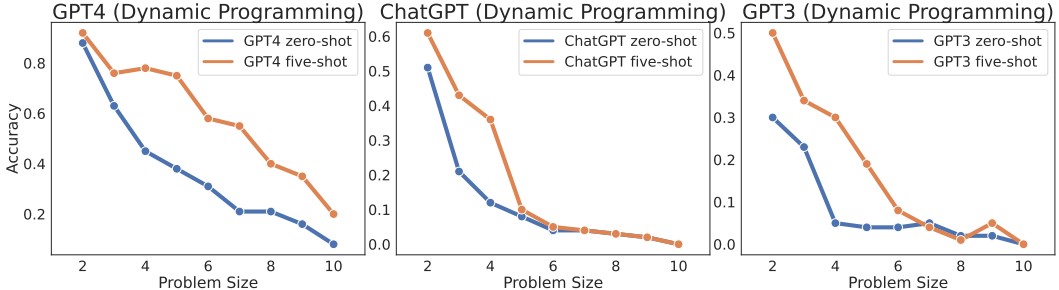

Figure 20: **Zero-shot** and **Few-shot accuracy** using **question-answer** pairs. Performance of GPT4, ChatGPT, and GPT3 on the **dynamic programming** task. LLaMA and FlanT5 results are near zero for all problem sizes.

## B.4   Limits of Transformers with Explicit Scratchpad Training

Figure 23, 24, 22 show the performance of GPT3 fine-tuned on different splits of the tasks using question-scratchpad pairs. Specifically, for the multiplication task, the model was fine-tuned on a range of multiplication problems, spanning from 1-digit by 1-digit multiplication to 3-digit by 2-digit multiplication.

As for the puzzle task, the model was fine-tuned on puzzles of sizes ranging from 2x2 to 4x4. Additionally, for the DP task, the model was fine-tuned on problems with a sequence length of 5. Furthermore, different data splits were considered, including variations based on the number of hours, number of properties, depth and width of the graph, and the number of digits in the multiplication output. On all tasks, we can see that the model fails to generalize to OOD data while achieving perfect accuracy on in-domain data, indicating that it cannot learn the underlying computational rules.

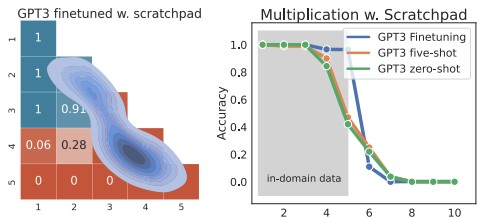

Figure 23: GPT3 finetuned exhaustively on task-specific data up to a certain problem size. In particular, we train on examples up to 3-digit by 2-digit multiplication (left) and on examples that have up to 5 digits in the output response (right). The **blue** region represents the in-distribution examples and the **red** region refers to OOD examples.

## B.5   Limits of Transformers with Explicit Scratchpad Prompting

Figure 25 shows the results. GPT-4 exhibits an increase in few-shot accuracy in most problem sizes when using question-scratchpad pairs of few-shot examples across the three tasks. While its performance surpasses that of zero-shot and few-shot with question-answer pairs, it tends to decline as the complexity of the tasks increases. The same applies for the rest of the models.

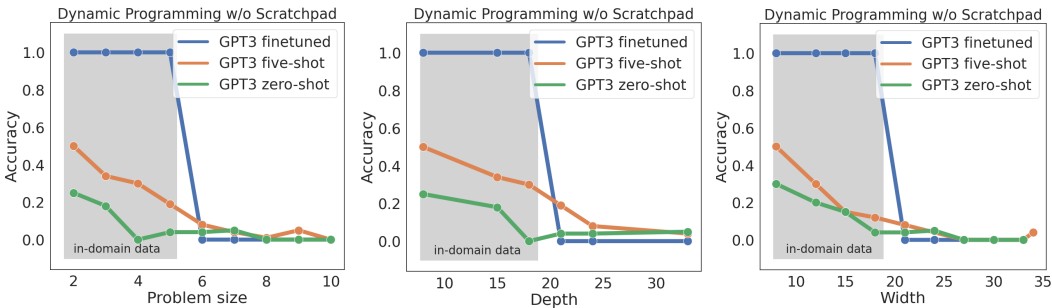

Figure 21: GPT3 finetuned on the **dynamic programming** task using **question-answer** pairs. We consider different data splits: problem size, depth, and width of the graph. Specifically, the model was trained with a problem size of 5, and the graph's depth and width were set to 18.

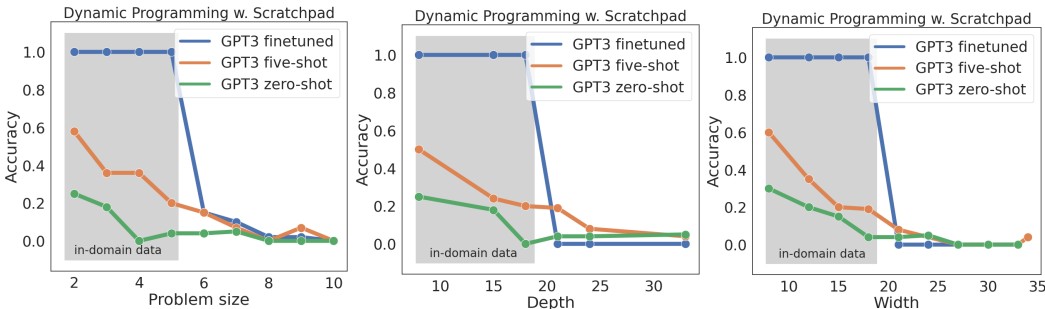

Figure 22: GPT3 finetuned on the **dynamic programming** task using **question-scratchpad** pairs. We consider different data splits: problem size, depth, and width of the graph. Specifically, the model was trained with a problem size of 5, and the graph's depth and width were set to 18.

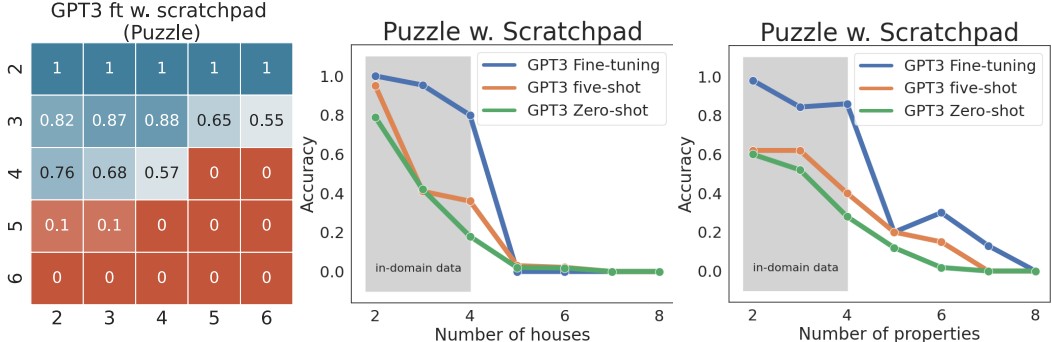

Figure 24: GPT3 finetuned on the puzzle task using **question-scratchpad** pairs. The training data consisted of puzzles of size 4x4, and the model was subsequently evaluated on larger puzzle sizes for OOD testing.

| Problem size | # examples | GPT3 Cost | |
| --- | --- | --- | --- |
| | | without scratchpad | with scratchpad |
| 1 x 1 | 81 | $0.12 | $7.44 |
| 2 x 1 | 810 | $1.28 | $74.4 |
| 2 x 2 | 8100 | $12.96 | $744 |
| 3 x 1 | 8100 | $12.96 | $744 |
| 3 x 2 | 81000 | $129.6 | $7440 |
| 3 x 3 | 810000 | $1296 | $74,404 |
| 4 x 1 | 81000 | $129.6 | $7440 |
| 4 x 2 | 810000 | $1296 | $74,404 |
| 4 x 3 | 8100000 | $12,960 | $744,040 |
| 4 x 4 | 81000000 | $129,600 | $7,440,400 |
| 5 x 1 | 810000 | $1296 | $74,404 |
| 5 x 2 | 8100000 | $12,960 | $744,040 |
| 5 x 3 | 81000000 | $129,600 | $7,440,400 |
| 5 x 4 | 810000000 | $1,296,000 | $70,440,400 |
| 5 x 5 | 8100000000 | $12,960,000 | $700,440,400 |

Table 1: Finetuning cost of GPT3 model on the multiplication data.

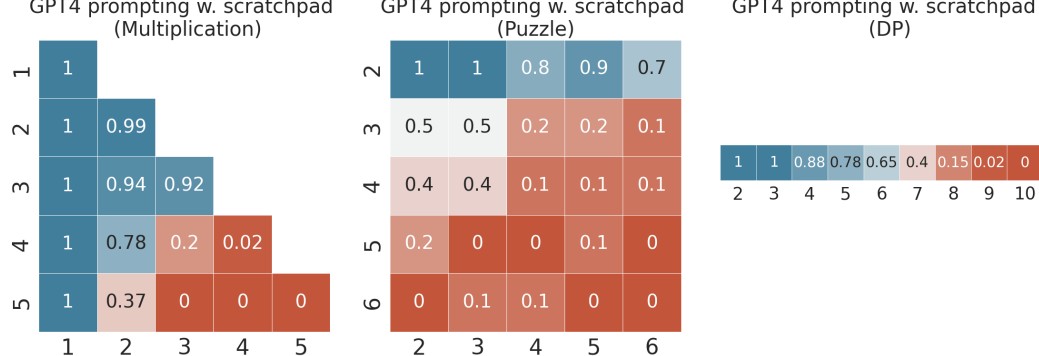

Figure 25: GPT4 few-shot accuracy with question-scratchpad pairs on the 3 tasks. The performance improves for most problem sizes compared to zero-shot performance and few-shot using question-answer pairs but degrades to zero as the complexity increases.

# C  Surface Patterns

## C.1  Relative Information Gain Predictions for Multiplication

| Input variable | Output variable | Relative Information Gain 2x2 | 3x3 | 4x4 | 5x5 |
|---|---|---|---|---|---|
| $x_n$ | $z_{2n}$ | 0.223 | 0.223 | 0.223 | 0.223 |
| $y_n$ | $z_{2n}$ | 0.223 | 0.223 | 0.223 | 0.223 |
| $x_1$ | $z_1$ | 0.198 | 0.199 | 0.199 | 0.199 |
| $y_1$ | $z_1$ | 0.198 | 0.199 | 0.199 | 0.199 |
| $x_n\ y_n$ | $z_{2n}$ | 1.000 | 1.000 | 1.000 | 1.000 |
| $x_{n-1}\ x_n$ | $z_{2n}$ | 0.223 | 0.223 | 0.223 | 0.223 |
| $y_{n-1}\ y_n$ | $z_{2n}$ | 0.223 | 0.223 | 0.223 | 0.223 |
| $x_n\ y_n$ | $z_{2n-1}$ | 0.110 | 0.101 | 0.101 | 0.101 |
| $y_{n-1}\ y_n$ | $z_{2n-1}$ | 0.032 | 0.036 | 0.036 | 0.036 |
| $x_{n-1}\ x_n$ | $z_{2n-1}$ | 0.032 | 0.036 | 0.036 | 0.036 |
| $x_{n-1}\ y_{n-1}$ | $z_{2n-1}$ | 0.018 | 0.025 | 0.025 | 0.025 |
| $x_1\ y_1$ | $z_2$ | 0.099 | 0.088 | 0.088 | 0.088 |
| $x_2\ y_2$ | $z_2$ | 0.025 | 0.016 | 0.016 | 0.016 |
| $x_1\ y_1$ | $z_1$ | 0.788 | 0.792 | 0.793 | 0.793 |
| $y_1\ y_2$ | $z_1$ | 0.213 | 0.211 | 0.211 | 0.211 |
| $x_1\ x_2$ | $z_1$ | 0.213 | 0.211 | 0.211 | 0.211 |

Table 2: **Highest Relative Information Gain Elements and Pairs of Elements**, for multiplications between $x = (x_1, \ldots, x_n)$ and $y = (y_1, \ldots, y_n)$, with $2 \leq n \leq 5$. We define $z := x \cdot y$, which will always have size $2n$ (with possibly a leading zero). $z_{2n}$ denotes the least-significant digit of $z$, and $z_1$ denotes the left-most digit. Only (input, output) pairs above 0.01 are shown. Note that since multiplication is commutative, several pairs of input variables (e.g. $a_0$ and $b_0$) exhibit the same relative information gain.

## C.2  Empirical Surface Pattern Analysis for Multiplication with GPT4, ChatGPT and GPT3

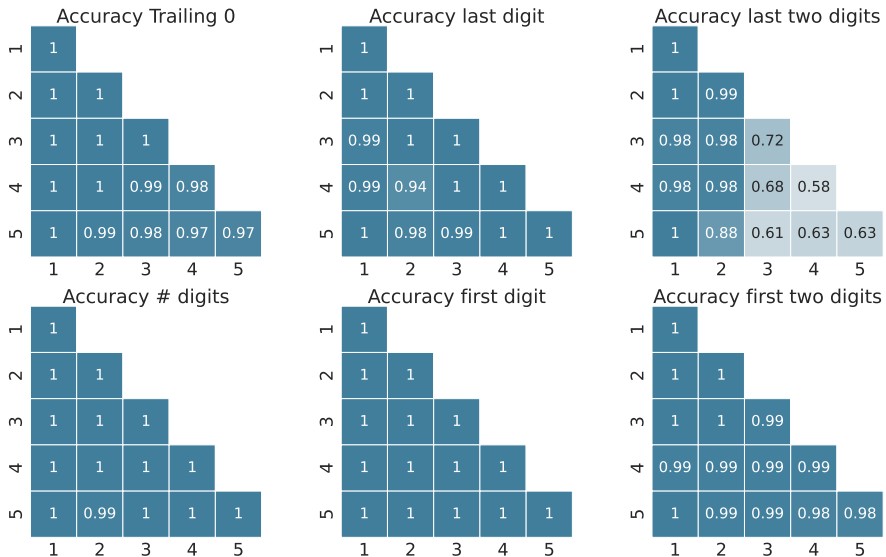

Figure 26: GPT4 zero-shot accuracy in predicting partially correct responses. This evidences surface pattern learning, since the accuracy of full answer prediction is significantly lower–and often near zero (see Figure 2). Specifically, 'accuracy trailing zeros' pertains to accurately predicting the number of zeros in the output number, which is known to be relatively easy to predict based on arithmetic calculations.

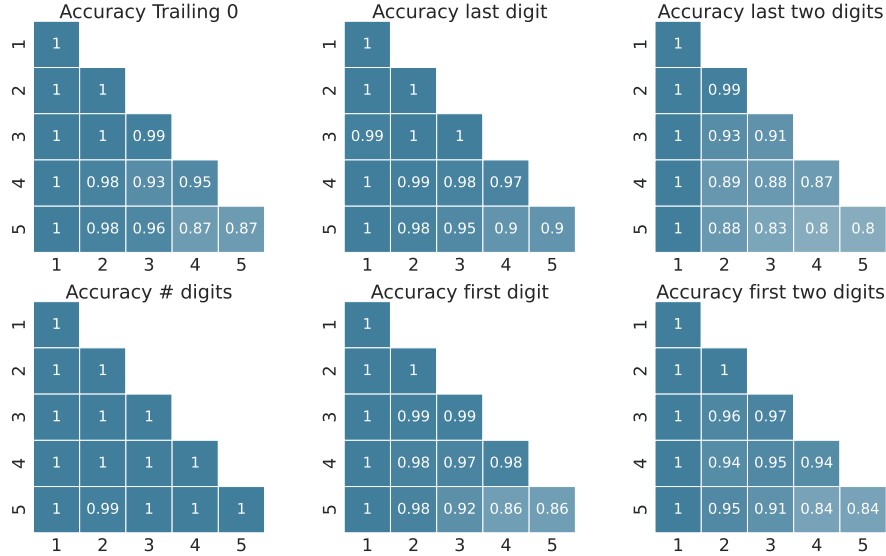

Figure 27: ChatGPT zero-shot accuracy in predicting partially correct responses. We observe the same trend for GPT3 predictions.

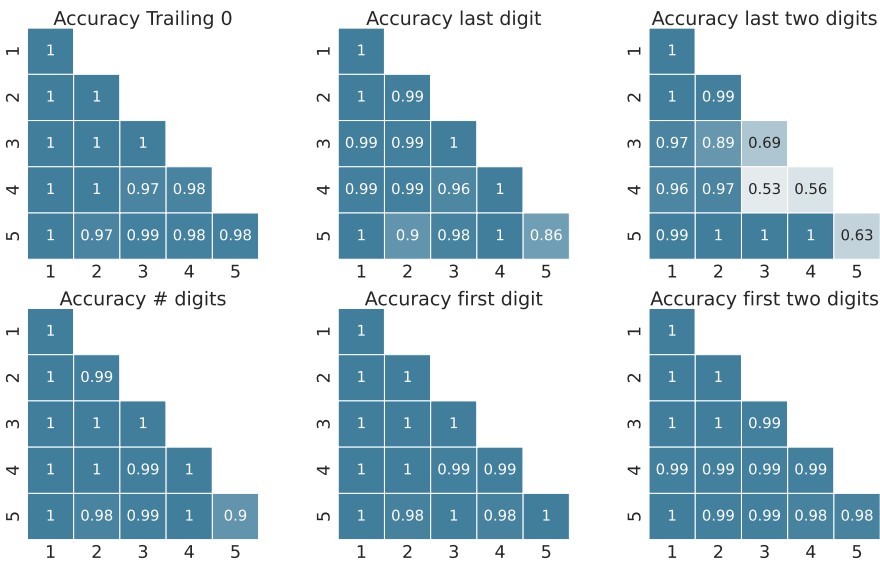

Figure 28: GPT4 five-shot accuracy in predicting partially correct responses. We observe the same trend for ChatGPT, GPT3 few-shot predictions.

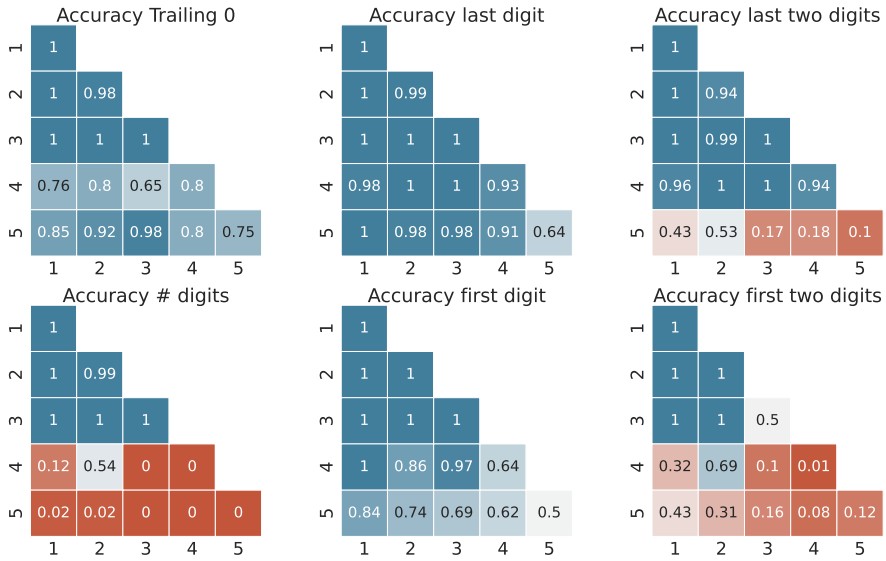

Figure 29: GPT3 finetuned on question-scratchpad pairs. Accuracy of predicting partially correct responses.

## C.3 Relative Information Gain Predictions for Dynamic Programming Task

Let $a_i$ be the $i$-th element of the input sequence, and let $o_i$ be the $i$-th element of the output sequence. As shown in Table 3, $a_i$ is a good predictor of $o_i$, and this is especially true for $a_1$ and $a_{n-1}$, the first and last elements of the sequence. This matches the task intuition, since one would never pick an element $a_i < 0$ and decrease the final sum (one may pick $a_i = 0$ if it makes a lexicographically smaller output sequence).

$a_i$ weakly helps to predict its neighbors. The only case of this behavior with RelativeIG>0.1 is at the start of the sequence, where the first element helps predict the value of the second. This again matches intuition, since a very high $a_1$ indicates that with high probability $o_2$ will not be selected for the final subsequence.

| | | Relative Information Gain for each problem size | | | | | | | | |
|---|---|---|---|---|---|---|---|---|---|---|
| Input variable | Output variable | 2 | 3 | 4 | 5 | 6 | 7 | 8 | 9 | 10 |
| $a_1$ | $o_2$ | 0.15 | 0.13 | 0.14 | 0.14 | 0.14 | 0.14 | 0.14 | 0.14 | 0.14 |
| $a_1$ | $o_1$ | 0.64 | 0.71 | 0.69 | 0.69 | 0.69 | 0.69 | 0.69 | 0.69 | 0.69 |
| $a_2$ | $o_2$ | 0.53 | 0.42 | 0.45 | 0.44 | 0.45 | 0.44 | 0.44 | 0.45 | 0.44 |
| $a_3$ | $o_3$ | | 0.64 | 0.49 | 0.53 | 0.52 | 0.52 | 0.52 | 0.52 | 0.52 |
| $a_4$ | $o_4$ | | | 0.60 | 0.46 | 0.50 | 0.49 | 0.49 | 0.49 | 0.49 |
| $a_5$ | $o_5$ | | | | 0.62 | 0.47 | 0.51 | 0.50 | 0.50 | 0.50 |
| $a_6$ | $o_6$ | | | | | 0.61 | 0.47 | 0.51 | 0.49 | 0.50 |
| $a_7$ | $o_7$ | | | | | | 0.61 | 0.47 | 0.51 | 0.50 |
| $a_8$ | $o_8$ | | | | | | | 0.61 | 0.47 | 0.51 |
| $a_9$ | $o_9$ | | | | | | | | 0.61 | 0.47 |
| $a_{10}$ | $o_{10}$ | | | | | | | | | 0.61 |
| $a_{n-1}$ | $o_{n-1}$ | | 0.64 | 0.60 | 0.62 | 0.61 | 0.61 | 0.61 | 0.61 | 0.61 |
| $a_{n-2}$ | $o_{n-2}$ | | | | 0.46 | 0.47 | 0.47 | 0.47 | 0.47 | 0.47 |
| $a_{n-3}$ | $o_{n-3}$ | | | | | | 0.51 | 0.51 | 0.51 | 0.51 |
| $a_{n-4}$ | $o_{n-4}$ | | | | | | | | 0.49 | 0.50 |

Table 3: **Highest Relative Information Gain Elements**, for DP problems of size $2 \leq n \leq 10$. We only show the (input, output) pairs where at least three problem sizes have RelativeIG>0, and at least one with RelativeIG>0.1. $a_{n-1}$ refers to the last element of the sequence, regardless of its actual id in the sequence.

Similar behaviors, but with higher relative information gains overall, are observed when analyzing triples of consecutive elements in the list. Table 4 shows that $o_i$ is highly predicted by $(a_{i-1}, a_i, a_{i+1})$. Moreover, $o_i$ is highly predicted by both $(a_{i-2}, a_{i-1}, a_i)$ and $(a_i, a_{i+1}, a_{i+2})$, with the former generally having higher scores than the latter. This again matches the task intuitions, since the value of the neighbors helps determine whether to select a number for the subsequence; and asking for the lexicographically smallest sequence biases the output subsequence to care more about the previous numbers rather than the following ones. We believe that this last point is the cause of the weakly predictive power of $(a_{i-3}, a_{i-2}, a_{i-1})$ to predict $o_i$; whereas $(a_{i+1}, a_{i+2}, a_{i+3})$ is not shown, since all the relative information gain values were below 0.1.

| Input variable | Output variable | Relative Information Gain for each problem size | | | | | | | |
| --- | --- | --- | --- | --- | --- | --- | --- | --- | --- |
| | | 3 | 4 | 5 | 6 | 7 | 8 | 9 | 10 |
| $a_{n-3}\,a_{n-2}\,a_{n-1}$ | $o_{n-1}$ | | | | | 0.95 | 0.95 | 0.95 | 0.95 |
| $a_{n-3}\,a_{n-2}\,a_{n-1}$ | $o_{n-2}$ | | | | | 0.87 | 0.87 | 0.87 | 0.87 |
| $a_{n-3}\,a_{n-2}\,a_{n-1}$ | $o_{n-3}$ | | | | | 0.64 | 0.64 | 0.64 | 0.64 |
| $a_1\,a_2\,a_3$ | $o_1$ | 1.00 | 0.96 | 0.97 | 0.97 | 0.97 | 0.97 | 0.97 | 0.97 |
| $a_1\,a_2\,a_3$ | $o_2$ | 1.00 | 0.91 | 0.92 | 0.91 | 0.92 | 0.91 | 0.92 | 0.91 |
| $a_2\,a_3\,a_4$ | $o_2$ | | 0.56 | 0.55 | 0.55 | 0.55 | 0.55 | 0.55 | 0.56 |
| $a_1\,a_2\,a_3$ | $o_3$ | 1.00 | 0.66 | 0.73 | 0.71 | 0.72 | 0.72 | 0.72 | 0.72 |
| $a_2\,a_3\,a_4$ | $o_3$ | | 0.86 | 0.77 | 0.78 | 0.78 | 0.78 | 0.78 | 0.78 |
| $a_3\,a_4\,a_5$ | $o_3$ | | | 0.67 | 0.66 | 0.66 | 0.66 | 0.66 | 0.66 |
| $a_2\,a_3\,a_4$ | $o_4$ | | 0.94 | 0.64 | 0.7 | 0.68 | 0.69 | 0.69 | 0.69 |
| $a_3\,a_4\,a_5$ | $o_4$ | | | 0.88 | 0.79 | 0.81 | 0.8 | 0.8 | 0.8 |
| $a_4\,a_5\,a_6$ | $o_4$ | | | | 0.63 | 0.62 | 0.62 | 0.62 | 0.62 |
| $a_3\,a_4\,a_5$ | $o_5$ | | | 0.95 | 0.65 | 0.71 | 0.69 | 0.7 | 0.7 |
| $a_4\,a_5\,a_6$ | $o_5$ | | | | 0.87 | 0.78 | 0.79 | 0.79 | 0.79 |
| $a_5\,a_6\,a_7$ | $o_5$ | | | | | 0.64 | 0.63 | 0.63 | 0.64 |
| $a_4\,a_5\,a_6$ | $o_6$ | | | | 0.94 | 0.64 | 0.71 | 0.69 | 0.7 |
| $a_5\,a_6\,a_7$ | $o_6$ | | | | | 0.87 | 0.78 | 0.8 | 0.8 |
| $a_6\,a_7\,a_8$ | $o_6$ | | | | | | 0.64 | 0.62 | 0.63 |
| $a_5\,a_6\,a_7$ | $o_7$ | | | | | 0.95 | 0.64 | 0.71 | 0.69 |
| $a_6\,a_7\,a_8$ | $o_7$ | | | | | | 0.87 | 0.78 | 0.8 |
| $a_6\,a_7\,a_8$ | $o_8$ | | | | | | 0.95 | 0.64 | 0.71 |
| $a_1\,a_2\,a_3$ | $o_4$ | | 0.12 | 0.1 | 0.11 | 0.11 | 0.11 | 0.11 | 0.11 |
| $a_2\,a_3\,a_4$ | $o_5$ | | | 0.1 | 0.09 | 0.1 | 0.09 | 0.1 | 0.1 |
| $a_3\,a_4\,a_5$ | $o_6$ | | | | 0.11 | 0.1 | 0.1 | 0.1 | 0.11 |
| $a_4\,a_5\,a_6$ | $o_7$ | | | | | 0.11 | 0.09 | 0.1 | 0.11 |
| $a_5\,a_6\,a_7$ | $o_8$ | | | | | | 0.11 | 0.09 | 0.11 |

Table 4: **Highest Relative Information Gain Contiguous Triples**, for DP problems of size $3 \leq n \leq 10$. We only show the (input, output) pairs where at least three problem sizes have RelativeIG>0, and at least one with RelativeIG>0.1. $a_{n-1}$ refers to the last element of the sequence, regardless of its actual id in the sequence.

## C.4 Empirical Surface Pattern Results for Dynamic Programming Task

We observe that all analyzed models match the Relative Information Gain prediction that $o_1$ (whether the first element goes into the output sequence or not) should be the easiest value to predict (see Figures 30, 31, and 32). However, since GPT3 often predicts shorter output sequences than the required size, the analysis of the predictive power of $o_{n-1}$ is only done for GPT4. In GPT4, we observe that $o_{n-1}$ is among the easiest values to predict as expected by Relative Information Gain.

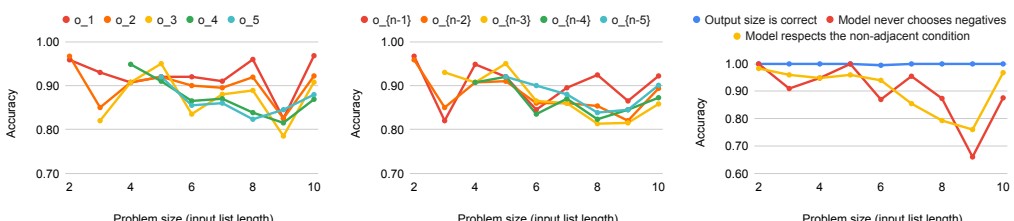

Figure 30: GPT4 five-shot with scratchpad accuracy in predicting output elements $o_i$ in the DP task. All $o_i$ are predicted with high accuracy with $o_1$ and $o_{n-1}$ being consistently among the highest. These observations go in line with the Relative Information Gain prediction.

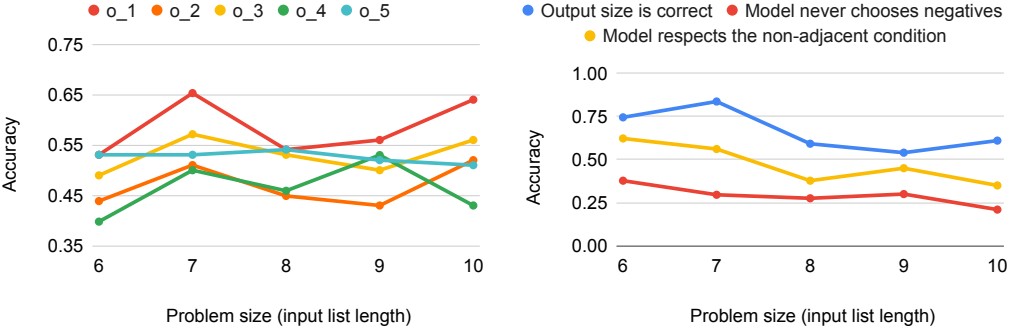

Figure 31: GPT3 few-shot without scratchpad accuracy in predicting output elements $o_i$ in the DP task. As predicted by Relative Information Gain, the model predicts $o_1$ correctly with the highest probability. However, because GPT3 often does not produce the correct output size, it hinders us from analyzing $o_{n-1}$.

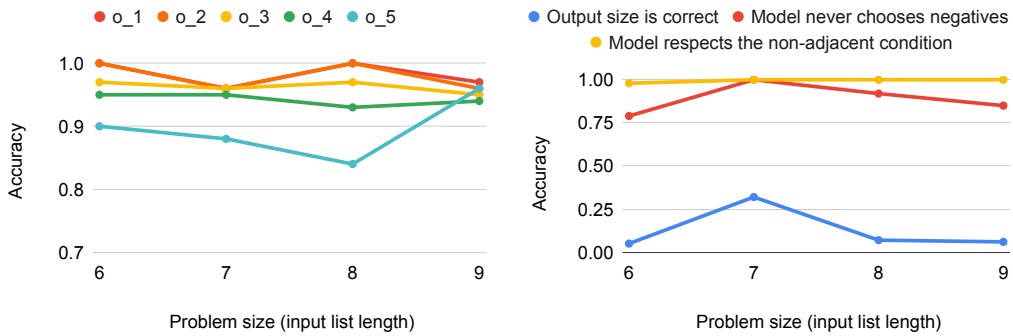

Figure 32: GPT3 fine-tuned without scratchpad accuracy in predicting output elements $o_i$ in the DP task. As predicted by Relative Information Gain, the model predicts $o_1$ correctly with the highest probability. However, because GPT3 often does not produce the correct output size, it hinders us from analyzing $o_{n-1}$.

# D Theoretical Results: Derivations

## D.1 Error accumulates with larger parallel applications of an estimated function (*width*)

Here we provide formal statements and derivations to Propositions 4.1 and 4.2 shown in the main paper. The mathematical framework used is a simplified representation of how multi-step reasoning works, showing two quintessential reasoning types: independent applications of the same step, or consecutive applications of the same step. We take an error estimation and accumulation perspective, since transformers are still being investigated from a theoretical standpoint.

**Proposition D.1.** *Let $f_n(\mathbf{x}) = h_n(g(\mathbf{x}, 1), g(\mathbf{x}, 2)), \ldots, g(\mathbf{x}, n))$. Let $\widehat{h}_n, \widehat{g}, \widehat{f}_n$ be estimators of $h_n, g, f_n$ respectively. Assume $\mathbb{P}(h_n = \widehat{h}_n) = 1$ and $\mathbb{P}(h_n(X) = h_n(Y) \mid X \neq Y) < c_n$, where $c_n < c$ for some constant $c \ll 1$ (i.e. $\widehat{h}_n$ perfectly estimates $h_n$, and $h_n$ is almost injective). If $\mathbb{P}(g \neq \widehat{g}) = \epsilon > 0$ and errors in $\widehat{g}$ are independent, $\liminf\limits_{n \to +\infty} \mathbb{P}(f_n \neq \widehat{f}_n) \geq 1 - c$.*

*Moreover, if $c_n \leq \beta \alpha^n$ for some some $\alpha \in (0, 1)$ and $\beta > 0$, then $\lim\limits_{n \to +\infty} \mathbb{P}(f_n \neq \widehat{f}_n) = 1$.*

*Proof.* For ease of writing, let $X_i = g(X, i)$ and $Y_i = \widehat{g}(X, i)$, and let $\mathbf{X} = (X_1, \ldots, X_n)$ and $\mathbf{Y} = (Y_1, \ldots, Y_n)$. We will compute some auxiliary probabilities, and then upper bound $\mathbb{P}(f = \widehat{f})$, to finally compute its limit.

$$
\begin{aligned}
\mathbb{P}(\mathbf{X} = \mathbf{Y}) &= \mathbb{P}(X_1 = Y_1, X_2 = Y_2, \ldots, X_n = Y_n) \\
&= \mathbb{P}(X_1 = Y_1) \cdot \mathbb{P}(X_2 = Y_2) \ldots \cdot \mathbb{P}(X_n = Y_n) = \mathbb{P}(g = \widehat{g})^n = (1 - \epsilon)^n \quad (2)
\end{aligned}
$$

Since by hypothesis we know $\mathbb{P}(h_n(\mathbf{Y}) = \widehat{h}_n(\mathbf{Y})) = 1$, we have that:

$$
\begin{aligned}
\mathbb{P}(h_n(\mathbf{X}) = \widehat{h}_n(\mathbf{Y}) \mid \mathbf{X} \neq \mathbf{Y}) &= \mathbb{P}(h_n(\mathbf{X}) = \widehat{h}_n(\mathbf{Y}) \cap h_n(\mathbf{Y}) = \widehat{h}_n(\mathbf{Y}) \mid \mathbf{X} \neq \mathbf{Y}) \\
&= \mathbb{P}(h_n(\mathbf{X}) = h_n(\mathbf{Y}) = \widehat{h}_n(\mathbf{Y}) \mid \mathbf{X} \neq \mathbf{Y}) \\
&\leq \mathbb{P}(h_n(\mathbf{X}) = h_n(\mathbf{Y}) \mid \mathbf{X} \neq \mathbf{Y}) \\
&< c_n \quad (3)
\end{aligned}
$$

We will now estimate $\mathbb{P}(f_n = \widehat{f}_n)$ using the law of total probability w.r.t. the event $\mathbf{X} = \mathbf{Y}$.

$$
\begin{aligned}
\mathbb{P}(f_n = \widehat{f}_n) &= \mathbb{P}(h_n(\mathbf{X}) = \widehat{h}_n(\mathbf{Y})) \\
&= \mathbb{P}(h_n(\mathbf{X}) = \widehat{h}_n(\mathbf{Y}) \mid \mathbf{X} = \mathbf{Y}) \cdot \mathbb{P}(\mathbf{X} = \mathbf{Y}) + \mathbb{P}(h_n(\mathbf{X}) = \widehat{h}_n(\mathbf{Y}) \mid \mathbf{X} \neq \mathbf{Y}) \cdot \mathbb{P}(\mathbf{X} \neq \mathbf{Y}) \\
&= \mathbb{P}(h_n(\mathbf{X}) = \widehat{h}_n(\mathbf{X})) \cdot \mathbb{P}(\mathbf{X} = \mathbf{Y}) + \mathbb{P}(h_n(\mathbf{X}) = \widehat{h}_n(\mathbf{Y}) \mid \mathbf{X} \neq \mathbf{Y}) \cdot (1 - \mathbb{P}(\mathbf{X} = \mathbf{Y})) \\
&= 1 \cdot (1 - \epsilon)^n + \mathbb{P}(h_n(\mathbf{X}) = \widehat{h}_n(\mathbf{Y}) \mid \mathbf{X} \neq \mathbf{Y}) \cdot (1 - (1 - \epsilon)^n) \quad \text{(using 2 and hypothesis)} \\
&< (1 - \epsilon)^n + c_n \cdot (1 - (1 - \epsilon)^n) \quad \text{(using 3)} \\
&< c_n + (1 - \epsilon)^n \cdot (1 - c_n)
\end{aligned}
$$

To conclude our proof, we will compute a lower bound for $\liminf_{n \to +\infty} \mathbb{P}(f_n \neq \widehat{f}_n)$. Note that since $c_n < c$ for all $n$, we know that $\mathbb{P}(f_n = \widehat{f}_n) < c + (1 - \epsilon)^n \cdot (1 - c)$. Then, $\mathbb{P}(f_n \neq \widehat{f}_n) > 1 - c - (1 - \epsilon)^n \cdot (1 - c)$. Since $1 - \epsilon \in [0, 1)$, $\lim\limits_{n \to +\infty} 1 - c - (1 - \epsilon)^n \cdot (1 - c) = 1 - c$. Thus,

$$
\liminf_{n \to +\infty} \mathbb{P}(f_n \neq \widehat{f}_n) \geq \liminf_{n \to +\infty} 1 - c - (1 - \epsilon)^n \cdot (1 - c) = 1 - c
$$

which concludes our proof.

**Note:** In the case where $c_n \leq \beta \alpha^n$, we can derive an even stronger conclusion. In this case, we can prove that $\lim\limits_{n \to +\infty} \mathbb{P}(f_n = \widehat{f}_n) = 1$. Recall that $\mathbb{P}(f_n = \widehat{f}_n) < \alpha \beta^n + (1 - \epsilon)^n \cdot (1 - \alpha \beta^n)$. Note that since $1 - \epsilon \in [0, 1)$ and $\alpha \in (0, 1)$, trivially $\lim\limits_{n \to +\infty} \beta \alpha^n + (1 - \epsilon)^n \cdot (1 - \beta \alpha^n) = 0$.

$$
0 \leq \liminf_{n \to +\infty} \mathbb{P}(f_n = \widehat{f}_n) \leq \limsup_{n \to +\infty} \mathbb{P}(f_n = \widehat{f}_n) \leq \limsup_{n \to +\infty} \beta \alpha^n + (1 - \epsilon)^n \cdot (1 - \beta \alpha^n) = 0
$$

Then, $\lim_{n \to +\infty} \mathbb{P}(f_n = \widehat{f}_n) = 0$ and we conclude $\lim_{n \to +\infty} \mathbb{P}(f_n \neq \widehat{f}_n) = 1$, assuming that $c_n \leq \beta \alpha^n$ for some adequate $\alpha, \beta$. $\qquad \square$

**Corollary D.1.** *Assume that a model $\mathcal{M}$ solves shifted addition perfectly, but it incorrectly solves **at least one** $m$ digit by 1 digit multiplication for some fixed $m$. Then, the probability that $\mathcal{M}$ will solve **any** $m$ digit by $n$ digit multiplication using the long-form multiplication algorithm tends to 0 when $n$ tends to infinity.*

*Proof.* Let $g = d \circ s$ define the base-10 multiplication between $m$-digit numbers $(x_1 x_2 \ldots x_m)$ and 1-digit numbers $(x_{m+j})$, where $s : \mathbb{Z}_{10}^{m+n} \times \mathbb{N} \to \mathbb{N} \times \mathbb{N}$ denotes the selection of the numbers to multiply and $d : \mathbb{N} \times \mathbb{Z}_{10} \to \mathbb{N}$ denotes the actual multiplication:

$$g := d \circ s$$
$$d(x, y) := x \cdot y$$
$$s([x_1, \ldots, x_m, x_{m+1}, \ldots, x_{m+n}], j) := (x_1 + \!\!+ x_2 + \!\!+ \ldots + \!\!+ x_m, \ x_{m+j})$$
$$\text{where } x_1 + \!\!+ x_2 + \!\!+ \ldots + \!\!+ x_m \text{ denotes concatenating digits } x_i$$

Let $h_n : \mathbb{N}^n \to \mathbb{N}$ describe the shifted addition used at the end of long-form multiplication to combine $n$ $m$-digit by 1-digit multiplications, and let $f_n : \mathbb{Z}_{10}^{m+n} \to \mathbb{N}$ describe the long-form multiplication of $m$-digit by $n$-digit numbers:

$$h_n(x_1, \ldots, x_n) := \sum_{i=1}^{n} x_i 10^{n-i}$$
$$f_n(\mathbf{x}) := h_n(g(\mathbf{x}, 1), g(\mathbf{x}, 2)), \ldots, g(\mathbf{x}, n))$$

By hypothesis, $\mathbb{P}(g \neq \widehat{g}) = \epsilon > 0$ and $\mathbb{P}(h_n = \widehat{h}_n) = 1$, where $\widehat{g}$ and $\widehat{h}_n$ denote estimators using model $\mathcal{M}$. It can be shown that $\mathbb{P}(h_n(X) = h_n(Y) \mid X \neq Y) < \beta \alpha^n$ for $\alpha = 0.1$ and $\beta = 10^m$. Using Lemma D.1, $\lim_{n \to +\infty} \mathbb{P}(f_n \neq \widehat{f}_n) = 1$, which concludes our proof.

$\qquad \square$

Note that Lemma D.1's proofs gives us empirical bounds once $\epsilon$ and $\alpha$ are approximated. Also **note that our definition of $g$ in the proof of Corollary D.1 highlights two possible sources of exponentially-accumulating error**: errors in the selection of the numbers to multiply $s$, and errors in the actual $m$-digit by 1-digit multiplication $d$.

## D.2 Error accumulates with larger iterative applications of an estimated function (*depth*)

**Proposition D.2.** *Let $f_n(\mathbf{x}) = g^n(\mathbf{x})$. Assume $\mathbb{P}(g(X) = \widehat{g}(Y) \mid X \neq Y) \leq c$ (i.e. recovering from a mistake due to the randomness of applying the estimator on an incorrect input has probability at most $c$). If $\mathbb{P}(g \neq \widehat{g}) = \epsilon > 0$ with $c + \epsilon < 1$, then $\liminf_{n \to +\infty} \mathbb{P}(f_n \neq \widehat{f}_n) \geq 1 - c/(c + \epsilon)$.*

*Proof.* We first derive a recursive upper bound using the law of total probability, and then prove a non-recursive upper bound by induction.

$$\begin{aligned}
s_n := \mathbb{P}(f_n = \widehat{f}_n) &= \mathbb{P}(g(g^{n-1}(Z)) = \widehat{g}(\widehat{g}^{n-1}(Z))) \\
&= \mathbb{P}(g(\mathbf{X}) = \widehat{g}(\mathbf{Y})) \quad \text{where } \mathbf{X} := g^{n-1}(Z) \text{ and } \mathbf{Y} := \widehat{g}^{n-1}(Z) \\
&= \mathbb{P}(g(\mathbf{X}) = \widehat{g}(\mathbf{Y}) \mid \mathbf{X} = \mathbf{Y}) \cdot \mathbb{P}(\mathbf{X} = \mathbf{Y}) + \mathbb{P}(g(\mathbf{X}) = \widehat{g}(\mathbf{Y}) \mid \mathbf{X} \neq \mathbf{Y}) \cdot \mathbb{P}(\mathbf{X} \neq \mathbf{Y}) \\
&= \mathbb{P}(g(\mathbf{X}) = \widehat{g}(\mathbf{X})) \cdot \mathbb{P}(\mathbf{X} = \mathbf{Y}) + \mathbb{P}(g(\mathbf{X}) = \widehat{g}(\mathbf{Y}) \mid \mathbf{X} \neq \mathbf{Y}) \cdot (1 - \mathbb{P}(\mathbf{X} = \mathbf{Y})) \\
&= \mathbb{P}(g(\mathbf{X}) = \widehat{g}(\mathbf{X})) \cdot s_{n-1} + \mathbb{P}(g(\mathbf{X}) = \widehat{g}(\mathbf{Y}) \mid \mathbf{X} \neq \mathbf{Y}) \cdot (1 - s_{n-1}) \\
&\leq (1 - \epsilon) \cdot s_{n-1} + c \cdot (1 - s_{n-1}) \\
&\leq (1 - \epsilon - c) \cdot s_{n-1} + c
\end{aligned}$$

We know $s_1 = (1 - \epsilon)$ since $s_1 = \mathbb{P}(f_1 = \widehat{f}_1) = \mathbb{P}(g = \widehat{g})$. Let $b := 1 - \epsilon - c$ for ease of writing. Then, we have

$$s_n \leq b \cdot s_{n-1} + c \qquad (4)$$

It can be easily shown by induction that $s_n \leq b^{n-1}(1 - \epsilon) + c \sum_{i=0}^{n-2} b^i$:

- The **base case** $n = 2$ is true since we know $s_2 \leq b \cdot s_1 + c$, and $b \cdot s_1 + c = b(1 - \epsilon) + c = b^{2-1}(1 - \epsilon) + c \sum_{i=0}^{2-2} b^i$, thus showing $s_2 \leq b^{2-1}(1 - \epsilon) + c \sum_{i=0}^{2-2} b^i$

- The **inductive step** yields directly using Equation 4,

$$s_n \leq b \cdot s_{n-1} + c$$

$$\leq b \cdot \left( b^{n-2}(1 - \epsilon) + c \sum_{i=0}^{n-3} b^i \right) + c \leq b^{n-1}(1 - \epsilon) + c \sum_{i=1}^{n-2} b^i + c \leq b^{n-1}(1 - \epsilon) + c \sum_{i=0}^{n-2} b^i$$

We can rewrite the geometric series $\sum_{i=0}^{n-2} b^i$ in its closed form $\frac{1 - b^{n-1}}{1 - b}$, and recalling $b := 1 - \epsilon - c$,

$$s_n \leq b^{n-1}(1 - \epsilon) + c \frac{1 - b^{n-1}}{1 - b} = b^{n-1}(1 - \epsilon) + c \frac{1 - b^{n-1}}{c + \epsilon}$$

$$= b^{n-1}(1 - \epsilon) + \frac{c}{c + \epsilon} - b^{n-1} \frac{c}{c + \epsilon}$$

$$= b^{n-1} \left( 1 - \epsilon - \frac{c}{c + \epsilon} \right) + \frac{c}{c + \epsilon}$$

Recalling that $s_n = \mathbb{P}(f_n = \widehat{f}_n)$, we compute the limit inferior of $\mathbb{P}(f_n \neq \widehat{f}_n) = 1 - s_n \geq 1 - b^{n-1}(1 - \epsilon - \frac{c}{c+\epsilon}) - \frac{c}{c+\epsilon}$.

$$\liminf_{n \to +\infty} \mathbb{P}(f_n \neq \widehat{f}_n) \geq \lim_{n \to +\infty} 1 - b^{n-1} \left( 1 - \epsilon - \frac{c}{c + \epsilon} \right) - \frac{c}{c + \epsilon} = 1 - \frac{c}{c + \epsilon}$$

that concludes our proof. $\qquad\square$

We can generalize the proof in Lemma 4.2 to tasks where there are potentially many valid reasoning chains with the following alternative state-transition framing.

**Lemma D.2.** *Let $S$ denote the set of all possible states a language model can generate, and let $z : S \to \{0, 1\}$ defines if a state is valid ($0 =$ invalid). Let $\widehat{g} : S \to \Pi(S)$ be a state-transition function representing a language model's probability distribution of generating each possible next state when attempting to perform a single reasoning step. Assume $\mathbb{P}(z(\widehat{g}(X)) = 1 \mid z(X) = 0) \leq c$ and $\mathbb{P}(z(\widehat{g}(X)) = 0 \mid z(X) = 1) = \epsilon > 0$ with $c + \epsilon < 1$. Then, $\liminf_{n \to +\infty} \mathbb{P}(z(\widehat{g}^n) = 0) = 1 - c/(c + \epsilon)$.*

If for task $T$ we know that all valid reasoning chains to arrive at a correct result have at least length $n$ (i.e., the equivalent of defining $f_n = g^n$ in Lemma D.1) then the probability of solving task $T$ correctly tends to at most $c/(c + \epsilon)$.

**Corollary D.3.** *The recursions for dynamic programming tasks, the $m$-by-1 digit multiplication, and the puzzle's elimination function are all tasks where there is a fixed reasoning step $g$ being repeatedly applied. Therefore, we can directly apply Proposition 4.2 to these tasks.*

*Proof.* Let's analyze the three tasks separately below.

$m$-**by**-1 **digit multiplication may be viewed as** $f^m(\mathbf{x})$   Let $x = (x_1, \ldots, x_m)$ be the $m$-digit number that we multiply by the 1-digit number $y$ ($0 \leq y < 10$). Let $z = (z_1, \ldots, z_{m+1})$ denote $z = x \cdot y$, which is guaranteed to have exactly $m + 1$ digits (with possibly leading zeros). We define $f$ as:

$$f(x_1, \ldots, x_m, y, i, c) := (x_1, \ldots, x_{i-1}, x_i', x_{i+1}, \ldots x_m, y, i - 1, c')$$

where $x_i' := (x_i \cdot y + c) \mod 10$ and $c' := \lfloor (x_i \cdot y + c)/10 \rfloor$. Note that $x_i' = z_{i+1}$ since $f$ is performing one step of the long-form multiplication algorithm.

Let the initial input be $\mathbf{x} := (x_1, \ldots, x_m, y, m, 0)$. Then, it can be easily shown that $f^m(\mathbf{x}) = (z_2, \ldots, z_{m+1}, y, 0, c)$. Since $c$ is the left-most carry, it is the leading digit of $z$, i.e. $c = z_1$ (possibly zero). Thus, the value of $z$ can be directly extracted from $f^m(\mathbf{x}) = (z_2, \ldots, z_{m+1}, y, 0, z_1)$.

**In the DP task, $dp$'s computation may be viewed as $f^{m-2}(x)$ for a list of size $m$** See §A.3.1 for details on the solution to this problem. We will use identical notation. Let $a_1, \ldots, a_m$ be an input list. Let $\mathbf{x} = (a_1, \ldots, a_{m-2}, a'_{m-1}, a'_m, m-2)$, where $a'_m := \max(a_m, 0)$ and $a'_{m-1} := \max(a_{m-1}, a_m, 0)$. Intuitively, this means that we have applied the first two steps of the $dp$ computation, and stored the results in $a'_{m-1}$ and $a'_m$. Let $f$ be a function representing the recursive computation of $dp_i$:

$$f(a_1, \ldots, a_i, a'_{i+1}, \ldots, a'_m, i) = (a_1, \ldots, a_{i-1}, a'_i, \ldots, a'_m, i-1)$$

where $a'_i := \max(a'_{i+1}, a_i + a'_{i+2}, 0)$.

Note that since $a'_{i+1}$ stores the value of $dp_{i+1}$ and $a'_{i+2}$ stores the value of $dp_{i+2}$, it can be easily shown that $f^{m-2}(\mathbf{x}) = (a'_1, \ldots, a'_m, 0) = (dp_1, \ldots, dp_m, 0)$. Therefore, $f^{m-2}$ computes all recursive values of $dp_i$ when given the base cases.

**In the DP task, the reconstruction of the desired subsequence given already computed $dp$ values may be viewed as $f^m(x)$ for an input list of size $m$.** This case is similar to the previous one. Let $r = (r_1, \ldots, r_m)$ be the result, where $r_i = 1$ if $a_i$ was selected for the desired subsequence, and $r_i = 2$ otherwise. Let $\mathbf{x} := (dp_1, \ldots, dp_m, 0, 0, a_1, \ldots, a_m, 1, 1)$. Let $f$ be defined as follows:

$$f(dp_1, \ldots, dp_m, 0, 0, a'_1, \ldots, a'_{i-1}, a_i, \ldots, a_m, i, u) = (dp_1, \ldots, dp_m, 0, 0, a'_1, \ldots, a'_i, a_{i+1}, \ldots, a_m, i+1, u')$$

where $a'_i := 2 - \mathbb{1}\{dp_i = a_i + dp_{i+2} \text{ and } u = 1\}$ and $u := 1 - \mathbb{1}\{dp_i = a_i + dp_{i+2} \text{ and } u = 1\}$. Intuitively, $a'_i$ stores whether the $i$-th element of the list should be selected for the final subsequence, assigning 1 if the element should be taken, and 2 otherwise (i.e., $a'_i = r_i$). Moreover, if the $i$-th element has been selected, we mark that the next item will not be available using $u'$. Therefore, $f$ performs one step of the final output reconstruction as defined in §A.3.1.

It can be easily shown that $f^m(\mathbf{x}) := (dp_1, \ldots, dp_m, 0, 0, a'_1, \ldots, a'_m, m+1, u') = (dp_1, \ldots, dp_m, 0, 0, r_1, \ldots, r_m, m+1, u')$. Note that the extra two elements in the input state allow lifting the special cases $m-1$ and $m$ in the solution shown in §A.3.1 without falling out of bounds.

**Solving the puzzle task may be seen as $f^m$ for some $m$, where $f$ is the elimination function** Let $c_1, \ldots, c_n$ be the list of clues, let $H$ be the number of houses, and let $A$ be a partially filled solution of size $K \times M$ as defined in §2.4. Each cell $A_{ij}$ can take $H+1$ values: the $H$ options for the cell and the value ø, implying this cell has not been filled. An elimination step $f$ may be defined as:

$$f(c_1, \ldots, c_n, A_{11}, \ldots A_{1M}, \ldots, A_{K1}, \ldots A_{KM}) = (c_1, \ldots, c_n, A'_{11}, \ldots A'_{1M}, \ldots, A'_{K1}, \ldots A'_{KM})$$

where $A'$ is also a partially filled matrix, with $A_{ij} = A'_{ij}$ for every $A_{ij} \neq$ ø and where $A'$ has at least one more filled cell.

Let $\mathbf{x} = (c_1, \ldots, c_n, E)$ where $E$ is an empty matrix of size $K \times M$ (all cell values of $E$ are ø).

Then, a full solution is computed as $f^m(\mathbf{x})$ for some value of $m$ that increases with the problem size. In contrast to other tasks, the value of $m$ is not fixed, and depends on the task instance, but using solvers we know that $m$ increases with problem size. $\qquad\square$

### D.3 Discussing $c \ll \epsilon$ in the context of Proposition 4.2

Note that in Proposition 4.2, if $c \ll \epsilon$ then $\liminf_{n \to +\infty} \mathbb{P}(f_n \neq \widehat{f}_n) \approx 1$. This is because assuming $\epsilon = m \cdot c$ for some $m > 0$, we have $1 - \frac{c}{c+\epsilon} = 1 - \frac{c}{c+m\cdot c} = 1 - \frac{1}{m+1} = \frac{m}{m+1}$, and $\frac{m}{m+1}$ is a monotonically increasing function for all $m > 0$ that tends to 1 when $m$ goes to infinity. Therefore, large $m$'s (or alternatively, $c \ll \epsilon$) imply $\frac{m}{m+1}$ will be close to 1.

It is reasonable to assume $c \ll \epsilon$ when $g$ has low collision, since $c$ represents the probability of the estimator $\widehat{g}(y)$ arriving at the correct output $g(x)$ by chance when given the wrong input $y \neq x$.

If $g$ is discrete, it can take $|\text{Im}(g)|$ values, where $|\text{Im}(g)|$ denotes the cardinal of the image space of $g$. Assuming approximately uniform errors, $c \approx \epsilon/|\text{Im}(g)|$, which in turn implies $c \ll \epsilon$ since $g$ being low collision implies $|\text{Im}(g)|$ is large.

If $g$ is continuous, under appropriate assumptions it seems plausible that we can prove that $c \approx 0$ (e.g. if errors are approximately uniform).

Summarizing both cases, if errors are approximately evenly distributed we obtain that $\liminf\limits_{n \to +\infty} \mathbb{P}(f_n \neq \widehat{f_n}) \approx 1$.

# E  Additional Literature and Societal Impact

## E.1  Additional Literature

**Iterated Functions**  The process of repeatedly applying a noisy single operation or function $f$ can be related to iterated random functions [24]. In this latter literature, the focus is usually on the contractive regime in which accrued errors can be kept under control, and the subsequent convergence guarantees (e.g., [23]). When $f$ is an affine transformation, the process falls simultaneously between two perspectives: time series [31] and dynamic programming and control [6]. We leverage the former to discuss the often explosive errors of $f^n$.

## E.2  Societal Impact Discussion

Our work on analyzing the limitations of current transformers in compositional tasks can have a positive societal impact in several ways. By shedding light on these limitations, we contribute to a deeper understanding of the capabilities and constraints of these models. This knowledge is essential for researchers, developers, and policymakers in making informed decisions regarding the application of transformers in various domains.

Understanding the limitations of transformers in compositional reasoning is crucial for developing more reliable and robust AI systems. By identifying these shortcomings, we can direct future research efforts toward addressing these limitations and developing models that exhibit improved performance in handling complex tasks requiring compositional reasoning.

We do not foresee any negative societal impacts, as our analysis aims to understand the reasons behind transformers' failures and successes, but does not introduce any new model or dataset that future work may leverage.

