# OpenReview forum: "Faith and Fate: Limits of Transformers on Compositionality"
_NeurIPS.cc/2023/Conference — NeurIPS 2023 spotlight_

### Official Review · Reviewer_jnqc · 2023-07-03

**Soundness:** 4 excellent
**Presentation:** 4 excellent
**Contribution:** 4 excellent
**Rating:** 7
**Confidence:** 4

**Summary:**

This paper studies the compositional reasoning capability of Transformer models on three kinds of tasks, namely multi-digit multiplication, logic grid puzzles, and a dynamic programming problem. The compositional complexity of a given task, or the number of reasoning steps, can be easily controlled by varying lengths of inputs. This paper formulates the reasoning steps as a computation graph and uses characteristics like lengths and widths of the computation graph to qualify the level of complexity. For experimental evaluation, this work evaluates several Transformer models (i.e., GPT3, ChatGPT, GPT-4) using zero-shot, few-shot, and finetuning techniques. This paper shows that in zero-shot and few-shot settings, all three Transformer models deteriorate sharply from nearly perfect to nearly zero when increasing compositional complexity. Furthermore, finetuning GPT-3 with or without scratchpads does not help GPT-3 generalize to unseen and more complicated task instances. To gain a deeper understanding of these failures, this work finds that 1) Transformer models are likely to give partially correct predictions on outputs that can be determined by a small set of input features, and 2) Transformer models are likely to rely on computation patterns seen in training data and generalize poorly when they are not sufficient for OOD tests. Finally, with reasonable assumptions, this work gives a theoretical justification for why any models fail to generalize longer compositional reasoning tasks. Informally, as long as every single step is not perfect, the error will happen for a sufficiently long reasoning chain.

**Strengths:**

- This work carefully selects three representative compositional tasks and introduces computation graphs to systematically quantify the level of complexity.
- State-of-the-art transformer models like GPT-3, ChatGPT, and GPT-4 are used in the experimental evaluations. The observed limits are likely to be universal for existing transformer models.
- Some new insights in analyzing the successes and failures of Transformer models are presented. Specifically, relative information gain seems a likely reason that Transformer models tend to predict partially correct answers; and linearized subgraph matching suggests Transformer models tend to capture spurious patterns in the training.
- Four interesting categories of errors are proposed and can effectively characterize the mistaken behaviors of Transformer models when the compositional complexity increases. The restoration error is particularly interesting, as it captures an often ignored problem -- the model could give correct answers but because of wrong reasons.
- Some simple but formal analyses are proposed to illustrate the theoretical limits of Transformer models.


**Weaknesses:**

- It is well-known that large language models like GPT-3, ChatGPT, and GPT-4 behave poorly on complicated reasoning and planning tasks.  So, most results of this work are somewhat expected.

- Only three tasks are evaluated, and only one specific algorithm is chosen for each task. Although the chosen algorithm seems straightforward, it is possible that the particular algorithm may not be appropriate for Transformer models to learn. It does not rule out that the Transformer models may be good at learning other less intuitive algorithms for the evaluated tasks.

- Finetuning is only performed on GPT-3. It is understandable that finetuning larger models like ChatGPT or GPT4 is computationally prohibitive for a small research group. But, from a scientific point of view, the observations on GPT-3 may not generalize to ChatGPT or GPT4, and the emergent capabilities of larger language models may overcome the limits observed on GPT-3.

- The claim of subsection 3.2.2 is a bit subjective. Linearized subgraph matching is a means proposed in this work to better understand Transformer models. It does shine some light on why Transformer models fail for the chosen tasks, however, whether that's indeed Transformers' internal dynamics is not quite clear.

- The theoretical justification seems oversimplified. It has nothing to do with the Transformer models.

**Questions:**

Even assuming the computation graph is unique for a specific problem instance, exponentially many different sequential steps could express the same computation. How many sequences are extracted for a problem instance?

The definition (line 213) of whether a full computation is seen during training is not quite clear, particularly what's the precise meaning of the sign $\equiv$ here?

Figure 5b shows that predictions with large graph depth (e.g., > 16) tend to be all correct and have zero frequencies. It seems the Transformer models do generalize to more complicated tasks without reusing seen computation patterns in the training. Is it a plotting issue? Or is this interpretation wrong for some reason?

**Limitations:**

The authors discussed potential limitations, and there is no (obvious) negative societal impact of this work.

---

> ### Author Rebuttal · Authors · 2023-08-09
>
> We thank reviewer jnqc for their encouraging and detailed feedback. We are glad they highlighted we provide **“new insights in analyzing the successes and failures of Transformers”** using state-of-the-art models, which implies that **“the observed limits are likely to be universal for existing transformer models”**; as well as **“interesting”** error categorization analysis, which **“captures an often ignored problem”**.
> Key clarifications are outlined below, and we respond inline.
>
> ### “It is well-known that large language models (...) behave poorly on complicated reasoning and planning tasks. So, most results of this work are somewhat expected.”
> As reviewer jnqc highlights in their review, **our work focuses on providing novel insights on why and how models fail through the use of computation graphs**, and not merely on observing Transformers’ failure rates for reasoning tasks. We only dedicate a single subsection $\S$3.1 to reporting Transformers’ failures. We would nonetheless like to emphasize that a fraction of our research community tends to **glorify** models like GPT-4 as possessing **superhumans** reasoning capabilities. Thus, including this report ensures a more balanced portrayal of LLMs. Moreover, in $\S$3.2.3, we show that Transformers’ successes correlate with having seen significant subgraphs of the full graph required during training, which could help explain the underlying reasoning behind some of the impressive generations reported in prior work (e.g. Bubeck et al., 2023).
>
>
> ### “Only three tasks are evaluated, and only one specific algorithm is chosen for each task. (...) It does not rule out that the Transformer models may be good at learning other less intuitive algorithms for the evaluated tasks.”
>
> Our focus is on evaluating how/if models can follow the straightforward algorithms described, and in $\S$3.1 we correlate accuracy with algorithmic difficulty using metrics such as width, depth, and average parallelism. We intentionally selected very diverse task+algorithm pairs, and consistently saw similar behaviors throughout $\S$3. This consistency suggests that it's unlikely that different algorithms would give radically different results, especially taking into account $\S$4’s theoretical justifications.
>
> Even if unlikely, we acknowledge that one can never rule out this possibility completely, and we will add it to our limitations. We emphasize that our work serves as a stepping stone, laying the groundwork for future investigations into these unresolved questions. This is particularly crucial in the current AI landscape.
>
> ### “Finetuning is only performed on GPT-3. (...) finetuning larger models like ChatGPT or GPT4 is computationally prohibitive (...) But, from a scientific point of view, the observations on GPT-3 may not generalize to ChatGPT or GPT4, and the emergent capabilities of larger language models may overcome the limits observed on GPT-3.”
>
> OpenAI does not currently allow ChatGPT or GPT4 finetuning, and we, unfortunately, have no official information even on whether ChatGPT is a larger LM than GPT3 or not. Nonetheless, we observe similar limitations (in the limit) on ChatGPT and GPT4 as those shown for GPT3 in the zero-shot and few-shot settings, which leads us to believe that we would see similar trends on the finetuning case as well. We encourage OpenAI to publicly share findings if these limitations do not hold with GPT4!
>
> ###  "Linearized subgraph matching is a means proposed in this work to better understand Transformer models (...) whether that's indeed Transformers' internal dynamics is not quite clear."
> In  $\S$3.2.2, we highlight a notable correlation between the frequency of subgraph data and task performance. This observation led us to hypothesize that models rely on pattern matching rather than general reasoning abilities to predict correct outputs. However, we acknowledge that our results only indicate a correlation, not a causal relationship. This limitation arises because our access to large language models is restricted to limited API calls, preventing us from examining their internal dynamics. In fact, we had actually addressed this concern in our initial submission's Limitations section ($\S$8).
>
> ### “The theoretical justification (...) has nothing to do with the Transformer models.”
> Please refer to the joint response for a detailed explanation.
>
> ### What does sign ≡ mean in line 213?
> We mean that these graphs are isomorphic, with also a correspondence in the labels assigned–that they are essentially the same graph. To enhance clarity, we'll consider using the symbol ≅ and ensure this is explicitly mentioned in the text.
>
> ### "In Figure 5b, why do predictions with large graph depth (e.g., > 16) tend to be all correct and have zero frequencies?"
> This comes as a byproduct of having virtually no subgraphs of that depth. For the dots reported for graph depth >= 16, we have $log_{10}(x)≈0.02$ per the logarithmic plot, showing that there are only ~2 subgraphs of that depth. Readers should focus on the differences between correct and incorrect final answers for smaller depths, where there are a vast number of subgraphs of that depth seen during training. We will modify the plots to only report points where there are at least 30 subgraphs of that depth.
>
> ### “Even assuming the computation graph is unique for a specific problem instance, exponentially many different sequential steps could express the same computation. How many sequences are extracted for a problem instance?”
> We agree that there are multiple valid linearizations for a given computation graph, i.e. any topological ordering would work. We maintain the order in which nodes are generated in the canonical solutions for each task, and report the scratchpad used in Appendix A.

---

> > ### Comment · Reviewer_jnqc · 2023-08-16
> > **Thank you for the detailed replies**
> >
> > The replies helped to clarify my confusion, which I guess other readers may also share, so I wish the authors will incorporate them into future revisions.

---

> > > ### Author Response · Authors · 2023-08-16
> > > **Thank you for your thoughtful feedback and for your response!**
> > >
> > > Thanks again for your thoughtful feedback and for your response! We are delighted that the clarifications were helpful, and we have already incorporated them in the revised version of our paper.

---

### Official Review · Reviewer_cZLE · 2023-07-03

**Soundness:** 3 good
**Presentation:** 4 excellent
**Contribution:** 3 good
**Rating:** 7
**Confidence:** 4

**Summary:**

This work studies the ability of LLMs for compositional tasks. They propose a formalization called computation graphs. With this formalization, they further propose three different tasks: multi-digit multiplication, Einstein’s puzzle, and a dynamic programming problem. They test multiple LLMs on these three tasks with different training and inference settings. By varying the depth and width of the graph, they show that LLMs do not do well when the scale of the problem is large. In their further analysis, they point out two problems of LLM models: relying on single-step shortcuts and relying on similar training subgraphs. They also conduct an error classification and notice many hints for memorization (e.g., restoration error). Finally, this work presents a theoretical analysis showing the effect of error propagation.

**Strengths:**

1.	This work presents a relatively comprehensive and detailed evaluation of the compositional abilities for LLMs. The empirical results contain multiple training settings, and results w/wo scratchpads.
2.	The three tasks designed in this work can be valuable for future work on compositional tasks.
3.	The error analyses in this work are detailed and insightful. I really like the error classification analysis, which shows a more detailed description about LLM behaviors, and provide concrete and quantitative evidence for memorization, restoration error and error propagation.


**Weaknesses:**

1.	While the theoretical derivation is interesting, I find the takeaways and the perspective to be slightly underwhelming. The main takeaway seems to be error propagation lead to bad results when the problem scale is sufficiently large. This is true, but this is also true for everything, and for both machines and humans. To me, the more interesting problem lies in that why even n is not very large, compositional problems still seem to be relatively hard. Additionally, I think the theoretical derivation applies to sampling but not to greedy decoding.
2.	The idea of using a computation graph to control and analyze the difficulty of composition is not novel, see related works in [1][2][3]. While I do feel that the approach and objective of this paper are different enough compared to previous works, there should still be a discussion about these works.
3.	Minor presentation issue: Some experimental setup details are hard to find. For example, I cannot find how large the finetuning datasets are. And some of the other important details (e.g., how you split the train/test for finetuning experiments) are in the appendix. I understand the space is tight in the main paper, but I would encourage the authors to add some details to the main paper so that the results can be interpreted more easily.



[1] Josef Valvoda, Naomi Saphra, Jonathan Rawski, Adina Williams, and Ryan Cotterell. Benchmarking Compositionality with Formal Languages. In COLING 2022.

[2] Keysers, Daniel, Nathanael Schärli, Nathan Scales, Hylke Buisman, Daniel Furrer, Sergii Kashubin, Nikola Momchev et al. Measuring Compositional Generalization: A Comprehensive Method on Realistic Data. In ICLR 2020.

[3] Ben Bogin, Shivanshu Gupta, and Jonathan Berant. Unobserved Local Structures Make Compositional Generalization Hard. In EMNLP 2022.


**Questions:**

1.	The prompt in the scratchpad seems to be carefully designed. How many different prompts have you tried? How do different prompts influence the findings?
2.	How do you compute the entropies used in the information gains?
3.	Just to confirm, when doing the question-answer training, is there no scratchpads?


**Limitations:**

This work provided a limitation section in Sec. 8 in the main paper and a societal impact statement in Appendix E.

---

> ### Author Rebuttal · Authors · 2023-08-09
>
> We thank reviewer cZLE for their helpful feedback. We appreciate that Reviewer cZLE found our evaluation study **“comprehensive”** and **“detailed”**, and the designed tasks **“valuable”** for future work. We are also thrilled that Reviewer cZLE found the error analysis to be **“detailed”** and **“insightful”**, providing **“concrete”** and **“quantitative”** evidence for LLM behaviors.
>
> The key clarification points are outlined below, grouped by theme.
>
> ### “The main takeaway seems to be that error propagation lead to bad results when the problem scale is sufficiently large. (...) this is true for everything, and for both machines and humans. To me, the more interesting problem lies in that why even n is not very large, compositional problems still seem to be relatively hard”
>
> Our main goal in the theoretical section is to provide further perspective on the presented empirical results, making explicit the different ways in which compounding stochastic errors affect final performance. The current form of the propositions aims to emphasize performance degradation in the limit, but we will change the statements to explicitly show the bounds we derivate: **probability of success decreases exponentially with the required level of compositionality, which helps explain high failure rates even with small values of n**.
>
> We agree that **propositions apply to any high-performant estimator of reasoning tasks**,  in particular Transformer models. In fact, we mention in $\S$5 that these propositions do apply more broadly, and we will expand upon this discussion in $\S$4 as well.
> We focus on out-of-the-box autoregressive Transformers, aligning with the scope of our experiments and the recent claims about their general compositional capabilities. By employing the propositions, we intend to inform future research directions as outlined in $\S$5, including intriguing avenues like self-correction.
> This is a point where we believe out-of-the-box Transformers and humans differ, making propositions potentially not applicable: humans can review and correct their computations, possibly making the hypothesis $P(\hat{g} != g) = \epsilon > 0$ not hold.
>
> ### “I think the theoretical derivation applies to sampling but not to greedy decoding”
> It applies in both cases! For the case of greedy decoding, $\hat{g}$ is deterministic and the proofs can be read as a combinatorial solution, where the probability would literally be the fraction of results with the desired condition.
>
> ### “How many different prompts have you tried? How do different prompts influence the findings?”
> We thank Reviewer cZLE for these great questions.  We experimented with several scratchpads to identify the optimal prompt that yielded the best results for all models, including GPT-4, ChatGPT, and GPT-3.  Notably, we observed that when the scratchpad primarily consists of mathematical symbols without sufficient natural language explanations for the math operations, the models tend to exhibit poorer performance. As a result, we took great care to provide detailed and natural sentence explanations for each reasoning step. For reference, please see Figure 8 in Appendix A.1, which showcases the scratchpad used for multiplication.
>
> ### “How do you compute the entropies used in the information gains?”
>
> We exhaustively compute the Shannon entropy for all possible input-output pairs for the specified problem sizes: in other words, and following $\S$2.3’s terminology, we enumerate (X_1, …, X_n, Y_1, ..., Y_m)’s sample space. We have made sure to clarify this in the paper. For faster computation, one could randomly sample input-output pairs when computing entropy--of course, assuming a large enough sample size for the dimensionality of the features being considered.
>
> ### “when doing the question-answer training, is there no scratchpads?”
> Yes, when training on question-answer pairs, the “answer” contains the final answer without the inclusion of the scratchpad.
>
> ### Discuss additional related works
> Thanks for bringing up additional related work! Please refer to our overall response for a more detailed discussion of them.
>
> ### Fix the presentation issue about the training data details.
> We apologize for this inconvenience. We have made sure to bring those details to the main paper to remove any ambiguity.

---

> > ### Comment · Reviewer_cZLE · 2023-08-18
> >
> > Thank the authors for providing additional insights and experiment details in the detailed response! As indicated by my original score, I already believe this is a solid paper worth acceptance, and all the responses are also helpful. I will keep my score unchanged, and encourage the authors to expand the response on the error propagation part in the final version. It will be an additional important contribution from this paper if it can provide insights on what properties of reasoning problems (in contrast to other general NLG problems) make it hard, and what properties of Transformers (or auto-regressive models) make it error-prone.

---

> > > ### Author Response · Authors · 2023-08-19
> > > **Thank you for your thoughtful feedback and for your response!**
> > >
> > > Dear Reviewer,
> > >
> > > Thanks again for your detailed and thoughtful comments. We're delighted that you think the paper is solid and worth acceptance.
> > > We will make sure to include your suggestions in the final revision.

---

> ### Comment · Area_Chair_vTWU · 2023-08-17
>
> Dear Reviewer,
>
> The author has posted their rebuttal, but you have not yet posted your response. Please post your thoughts after reading the rebuttal and other reviews as soon as possible. All reviewers are requested to post this after-rebuttal-response.

---

### Official Review · Reviewer_euRW · 2023-07-06

**Soundness:** 4 excellent
**Presentation:** 3 good
**Contribution:** 3 good
**Rating:** 7
**Confidence:** 5

**Summary:**

This paper contributes to the literature on analyzing the limitations of Transformer models in compositional tasks by presenting an analysis based on computation graphs. Based on these computation graphs, some interesting new observations are made, which I believe are worth publishing.

I found the paper a great read and very interesting. I have many comments/criticisms/questions below, but I would like to clarify that these are not attempts to attack the content, and more aiming for constructive criticisms for trying to make the paper more clear and situated in the literature, as I think the work is very solid.

Some typos/comments:
- Part of this work has already been studied in the past, and I think the authors should do a better job in differentiating the new insights from previously known parts. For example, I think it'd be great to discuss the new findings given: "Making transformers solve compositional tasks" (2021), "The devil is in the detail: Simple tricks improve systematic generalization of transformers" (2021), and "Grokking: Generalization beyond Overfitting on Small Algorithmic Datasets" (2022)
- Page 2: "This substantial gap suggests that systematic problem-solving capabilities do not emerge from maximum likelihood training" -> See the Grokking paper I mentioned above, as they show that maybe systematic problem-solving abilities can improve when training beyond the overfitting threshold.
- Section 3: As discussed in "Making transformers solve..." mentioned above, even if "vanilla" Transformers really struggle to solve compositional tasks, slight modifications to the architecture show significant gains. So, when only evaluating variants of the GPT architecture, it seems like conclusions are limited to GPT-like models, and not to "Transformers" in general, as small architectural decisions can have big impact in these types of tasks. For example GPT models being decoder-only models, they employ causal attention, where as encoder-decoder models would be able to deploy bidirectional attention to the input problem, which can give very different inductive biases. Carefully clarifying the extent of the conclusions would be better.
- Section 3.1: "suggesting that systematic problem-solving capabilities do not emerge via exhaustive training on task-specific data." -> Again, I'd like to bring up the Grokking paper results, and I wonder if the authors could comment on the implications for their results here.
- Section 3.1: Although results are very interesting, I think the authors over-generalize the conclusions from experiments with one particular model pretrained in one particular way. I think it might be good to constrain conclusions of results to decoder-only-GPT-style-pretrained-Transformer models.
- Propositon 4.1: this has very little to do with Transformers, but with any method that approximates steps. And btw, this proposition applies to humans as well, as the probability that a human makes a long calculation correct decreases with the length of the calculation due to the growing probability that at least one of the steps is wrong. So, again, I think that clarifying the extent of the conclusions in the paper needs some work, whereas above I was pointing out that experiments show failure cases for GPT-style models and you are overgeneralizing to Transformers, here you are showing a limitation of sequential prediction models, and too narrowly just blaming Transformers. Moreover, I see this proposition very related to the classic error compounding in the literature of imitation learning (e.g., see "Efficient Reductions for Imitation Learning", 2010), and usual solutions, e.g. Dagger, rely on training models to recover from mistakes, mitigating the problem. It'd be great to comment on the relation if the authors see it appropriate.

**Strengths:**

Even if the weaknesses of Transformers in compositional tasks have been studied many times in the recent past, this paper brings an interesting new perspective based on computation graphs that allows for a more finegrained analysis than previous work.

**Weaknesses:**

Perhaps the two main weaknesses (fixable) are that (1) conclusions seem to be overgeneralizing to the whole class of Transformer models, when only one instance of them was used for evaluation, and (2) it seems that some of the most relevant pieces of work in the literature were not discussed, making it harder to see what are the new pieces of knowledge that this paper is bringing compared to what was already known. Both of them fixable with small modifications to the manuscript though.

**Questions:**

- In Section 3.2.3: How were these errors analyzed? Was the text output of the models parsed to find the intermediate steps? Assuming many of these are in natural language, how accurately were you able to recover these intermediate steps?
- Concerning memorization/shortcuts: We humans use memorization and shortcuts all the time for performing some of these operations (e.g., we know that X times 0 is 0, no matter how many digits does X have, and hence we will skip all the computation graph, etc.). The key it seems to me is to know when these shortcuts are applicable, and when they are dangerous. Did you see any case of the former (correct application of shortcuts)? (apologies if this is explained somewhere in the paper or appendix, but I was unable to find it).
- As a curiosity, where does the "faith and fate" from the title come from, it might be a lack of familiarity with some English idioms, but I am struggling to make the connection with the paper.


**Limitations:**

As acknowledged by the authors in their own section, the main limitation is that experiments were conducted with just GPT models, and it's unclear if we can generalize from those, given these are all the same Transformer architecture, and pretrained in a particular way.

---

> ### Author Rebuttal · Authors · 2023-08-08
>
> We thank Reviewer euRW for their comprehensive, insightful, and constructive feedback! We’re thrilled to hear that they found our paper **“a great read”**, **“very interesting”**, containing **“interesting new observations”** and **“worth publishing”**.
>
> ### On Grokking
> We thank Review euRW for bringing the grokking phenomenon to our attention! We are aware of this behavior and had already conducted experiments for the multiplication task. We observed no grokking behavior, and have already incorporated the new results and discussion: see Figure 1 in the additional page for results. We will also discuss the relevant literature in the Related Work section.
> Following Murty et al., 2023, we trained GPT3 with question-answer pairs for 420K steps, far exceeding the point at which in-domain accuracy plateaus (~90K steps). This training duration is equivalent to 60 epochs and **costed 50,000 USD**. We also trained GPT3 with question-scratchpad pairs for 30K steps, **costing 40,000 USD**. Our findings reveal that there is no improvement in generalization beyond the overfitting point, even after extensive training periods.
>
> **We postulate that the extent to which generalization improves beyond overfitting is intricately linked to the inherent complexity of the task.** First, as suggested by Nanda et al. 2022, grokking may emerge as the model’s regularization encourages generalization even beyond the point of overfitting. Second, as concluded by Liu et al 2022, generalization during grokking corresponds with the emergence of well-structured representations. However, the model might struggle to learn well-structured representations from the training data, leading to a failure in achieving grokking. Therefore, we suspect that grokking might not necessarily arise in all tasks. **We speculate that the inherent complexity of the multiplication task makes it significantly more challenging to learn a well-structured representation.**
>
> Future work is needed to understand grokking better, and as we mention in L319-326, we invite those equipped with much more substantial funding to investigate this further, as we couldn’t afford to spend any more than we have. It is important to note that even if some grokking were to happen through more prolonged training, such an approach would be inefficient and not scalable to more complex tasks.
>
> ### “Conclusions are limited to GPT models, and not to "Transformers" in general” E.g. encoder-decoder models. Clarify the extent of the conclusions."
> Our primary focus is to understand the performance of the most powerful general-purpose models available to date, hence the emphasis on GPT-like models.  In our initial submission, we had already included explorations of other models–FlanT5-11B and LLaMA-13B–in the appendix. We will add explicit pointers to these analyses in the main text. FlanT5, an encoder-decoder model, exhibits performance trends similar to the decoder-only models. Notably, we observe a decline in performance as the task complexity increases regardless of the model architecture, as illustrated in Figure 14, 15, 16. For more in-depth details, please refer to these figures.
>
> ### “Proposition 4.1: this has little to do with Transformers, but with any method that approximates steps. This proposition applies to humans as well (...)”
> Our main goal in the theoretical section is to provide further perspective on the presented empirical results, making explicit the ways in which compounding stochastic errors affect final performance. We talk about out-of-the-box autoregressive transformers to align with experiments, and because of recent claims of their reasoning skills. We agree that propositions apply to any high-performant estimator of reasoning tasks, as we allude in $\S$5: we will expand upon the discussion in $\S$4.
> Propositions potentially do not apply to humans, who can review and correct their computations, possibly making the hypothesis $P(\hat{g} != g) = \epsilon > 0$ not hold. This would also be the case when pairing Transformers with inference-time algorithms–a direction that our propositions enable us to discuss in $\S$5.
>
> ### Proposition 4.1 is “very related to the classic error compounding in the literature of imitation learning (...). It'd be great to comment on the relation”
> Thanks for bringing this to our attention! We will mention this in the discussion, as the analogy with imitation learning may spark future work on mitigating these issues in LLMs through novel algorithms in line with e.g. DAgger.
>
> ### How were errors analyzed? Parsing scratchpad?
> The models generate scratchpads using 2 approaches. 1) "in-context learning": prompting with 5 examples of question-srcatchpad pairs. 2) "finetuning on the question-scratchpad pairs". We build a computation graph out of the generated scratchpad via string regex parsing. Since the scatchpad language is templated (see Figure 8 in Appendix), our parsing accuracy is near perfect. This accuracy was confirmed through manual inspection of a randomly selected small set.
> To analyze errors, we compared the generated computation graph to the ground truth computation graph, and categorized each model
> generated step to four categories (fully correct, local error, propagation error, restoration error) as described in $\S$3.2.3.
>
> ### Concerning shortcuts
> We observed that, for problems with low complexity, Transformers could provide correct answers by relying on learned surface patterns, like "X times 0 is 0". In general, Transformers appear to struggle in discerning when to employ shortcut learning versus using the computation graph. Shortcut learning can be a speedy solution,  but it becomes brittle when machines lack the knowledge of when to use it.
>
> ### "faith and fate" meaning
> We wanted to allude to the faith (part of) the community has in Transformers’ general reasoning capabilities and contrast it with the fate of them failing when tested in the limit. We also think it is a nice alliteration!

---

> ### Comment · Area_Chair_vTWU · 2023-08-17
>
> Dear Reviewer,
>
> The author has posted their rebuttal, but you have not yet posted your response. Please post your thoughts after reading the rebuttal and other reviews as soon as possible. All reviewers are requested to post this after-rebuttal-response.

---

### Official Review · Reviewer_fn75 · 2023-07-06

**Soundness:** 3 good
**Presentation:** 4 excellent
**Contribution:** 3 good
**Rating:** 6
**Confidence:** 3

**Summary:**

This paper develops empirical and theoretical studies on the limitations of Transformers for solving compositional tasks. It defines computation graphs and quantifies compositional complexity with the graph metrics and uses relative information gain to predict surface patterns learned by Transformer. It conducts extensive experiments on three representative compositional tasks and finds that Transformers rely on pattern matching and exhibits poor generalization in compositionality.

**Strengths:**

This paper conducts extensive experiments and profound analysis. The idea of quantifying compositional complexity with computational graphs is inspiring.

**Weaknesses:**

The empirical findings and the driven conclusions are quite trivial.

**Questions:**

1. Based on the findings, what could be a possible solution for improving Transformers in multi-step compositional problems?
2. Can the computational graph definition and graph metrics be generalized to other compositional tasks, for example geometry problems?


**Limitations:**

The paper lacks methodologies or instructions for improving the model based on the obtained empirical findings and driven conclusions.

---

> ### Author Rebuttal · Authors · 2023-08-08
>
> We thank reviewer fn75 for their feedback. We are excited that reviewer fn75 stated our paper includes **“extensive experiments”** and **“profound analysis”** and found the idea of using computational graphs **“inspiring”**. As a result, we would like to seek clarification on some conflicting comments made by the reviewer.
>
> ### “The empirical findings and the driven conclusions are quite trivial”
>
> We would appreciate it if reviewer fn75 could clarify their reasons behind this comment. If Reviewer fn75 meant that Transformers are expected to fail on compositional tasks with increasing complexity, we would like to highlight the fact that there is a great deal of hype in the field around the capabilities of Transformers to the point that some people believe they can handle *any* task effortlessly. In this context, we believe that our findings are critical and timely to address potential hypes and misconceptions.
>
> However, going beyond demonstrating that Transformers struggle with compositional tasks (to which we only dedicate a single subsection $\S$3.1), more importantly, our findings and conclusions provide novel insights and a nuanced understanding of **why and when** Transformers succeed and struggle. To the best of our knowledge, no other research paper to date has tackled this question to the scale and depth to which this paper investigates. Moreover, we explore whether pushing Transformers to their limits can lead to complete mastery of tasks.
>
>
> We'd like to highlight the following valuable insights and contributions of our work:
>
> * **Formalization of tasks as computational graphs ($\S$2):**  This approach allowed us to provide a systematic and acute explanation of Transformers failures and successes. By quantifying compositional complexity using graph metrics, we gained a deeper understanding of their pattern matching behavior and could analyze errors at each reasoning step.
>
> * **Comprehensive analysis from multiple perspectives ($\S$3.2):** We used information gain ($\S$3.1) to explain instances where Transformers partially succeeded, performed subgraph matching analysis ($\S$3.2) to reveal Transformers' reliance on pattern matching, and carried out fine-grained error analysis ($\S$3.3) to categorize and illustrate various types of reasoning failures.
>
> * **Theoretical insights on Transformers' failure with increasing graph complexity ($\S$4):** We provide mathematical arguments that Transformers' performance struggles with problems that require increasingly larger iterative applications of a function (depth) and larger parallelism (width). These proofs conclusively indicate that Transformers, in their current state, can easily fail with highly complex compositional tasks.
>
> ### “The paper lacks methodologies or instructions for improving the model based on the obtained empirical findings”
>
> Enhancing Transformers' capabilities in compositional tasks **falls outside the scope of our work**. We would like to reiterate that our main goal is to explore Transformers' limits and gain a comprehensive understanding of the factors contributing to their success and failure. We've nonetheless provided a discussion ($\S$5) in our initial submission on promising avenues to improve performance based on our theoretical results. For example, augmenting Transformers with planning modules as well as using refinement methods, that can iteratively improve their generations. We believe that these possible avenues for improvement can be valuable directions for future research.
>
> ### “Can the computational graph definition and graph metrics be generalized to other compositional tasks, for example geometry problems?”
> Definitely! Any task for which a deterministic executable program can provide a solution is one where computational graphs can be directly applied. This includes computational geometry problems, with famous examples such as convex hull, Voronoi diagram generation, closest pair of points, polygon triangulation, among others. Additionally, other problems such as graph algorithms like BFS or Dijkstra, as well as sorting algorithms can be framed as computational graphs. Tasks with low computational complexity (e.g. O(n log n) for convex hull) are great candidates for scratchpad generation since scratchpad length correlates with the algorithm’s complexity.
>
> In conclusion, we hope our clarifications help to emphasize the significance of our paper's findings and the substantial contribution it makes. We are happy to address any additional concerns.

---

> ### Comment · Area_Chair_vTWU · 2023-08-17
>
> Dear Reviewer,
>
> The author has posted their rebuttal, but you have not yet posted your response. Please post your thoughts after reading the rebuttal and other reviews as soon as possible. All reviewers are requested to post this after-rebuttal-response.

---

### Author Rebuttal · Authors · 2023-08-09

## Overall Response
### Discuss Additional Related Work

We appreciate reviewer euRW and reviewer cZLE for bringing additional relevant works to our attention! Given the page limit constraint, we were unable to discuss those works in our initial submission. However, we will incorporate them into the related work section along with other works in the same vein in the revision. While relevant, these works have different goals from our study:

The following works [1, 2, 3] focus primarily on how to enhance LLMs performance on compositional problems. In contrast, we aim to investigate the **fundamental limits** of Transformers in achieving full mastery of the task by pushing the models’ limit in terms of training and prompting. Our primary goal is to determine **whether and why** the most powerful LLMs, such as GPT-4 and ChatGPT, can consistently achieve perfect performance in both in-domain and out-of-domain settings as the complexity of tasks increases. We offer novel insights into the intricacies involved, shedding light on the reasons why achieving full mastery on compositional problems is inherently challenging.

The works [4,5] propose novel methods to construct compositionality benchmarks and subsequently evaluate LLM behavior against these benchmarks. The work [6] discusses that improvement in generalization can happen well past the point of overfitting through a process called "grokking". However, these findings are limited to "small transformer model" and "small algorithmically generated datasets". It remains unclear whether the conclusion would generalize to LLMs dealing with complex tasks. We conducted additional experiments to investigate this phenomenon in our setting, please see the response to R2 for more details and the attached page.


### Expanding characterization of our theoretical results: we agree they apply to more agents than just Transformers!

Our main goal in the theoretical section is to provide further perspective on the presented empirical results, making explicit the different ways in which compounding stochastic errors affect final performance. We agree that **propositions apply to any high-performant estimator of reasoning tasks, and we mention in $\S$5 that they apply more broadly**: we will expand upon this discussion and mention this in $\S$4 as well. We focus on out-of-the-box autoregressive Transformers, aligning with the scope of our experiments and the recent claims about their general compositional capabilities. By employing the propositions, we intend to inform future research directions as outlined in $\S$5,  such as augmenting models with self-correction modules.

The current form of the propositions aims to emphasize performance degradation in the limit, but we will change the statements to explicitly show the bounds we derivate: **the probability of success decreases exponentially with the required level of compositionality**, which helps explain high failure rates even with small values of n. Providing *specific bounds for transformers based on their specific architecture is an interesting future work direction*.


**Bibliography**

[1] Ontanón, Santiago, Joshua Ainslie, Vaclav Cvicek, and Zachary Fisher. Making Transformers Solve Compositional Tasks. ACL 2022.

[2] Csordás, Róbert, Kazuki Irie, and Jürgen Schmidhuber. The Devil is in the Detail: Simple Tricks Improve Systematic Generalization of Transformers. EMNLP 2021.

[3] Bogin, Ben, Shivanshu Gupta, and Jonathan Berant. Unobserved Local Structures Make Compositional Generalization Hard. In EMNLP 2022.

[4] Valvoda, Josef, Naomi Saphra, Jonathan Rawski, Adina Williams, and Ryan Cotterell. Benchmarking Compositionality with Formal Languages. In COLING 2022.

[5] Keysers, Daniel, Nathanael Schärli, Nathan Scales, Hylke Buisman, Daniel Furrer, Sergii Kashubin, Nikola Momchev et al. Measuring Compositional Generalization: A Comprehensive Method on Realistic Data. In ICLR 2020.

[6] Power, Alethea, Yuri Burda, Harrison Edwards, Igor Babuschkin and Vedant Misra. “Grokking: Generalization Beyond Overfitting on Small Algorithmic Datasets.” arXiv preprint arXiv:2201.02177 (2022).

[7] Nanda, Neel, Lawrence Chan, Tom Liberum, Jess Smith, and Jacob Steinhardt. "Progress measures for grokking via mechanistic interpretability." In ICLR 2022.

[8] Liu, Ziming, Ouail Kitouni, Niklas S. Nolte, Eric Michaud, Max Tegmark, and Mike Williams. "Towards understanding grokking: An effective theory of representation learning."  In NeurIPS 2022.

[9] Murty, Shikhar, Pratyusha Sharma, Jacob Andreas, and Christopher D. Manning. Grokking of Hierarchical Structure in Vanilla Transformers.  In ACL 2023

---

### Decision · Program_Chairs · 2023-09-21

**Decision:**

Accept (spotlight)

**Comment:**

The paper investigates the limitations of Transformer-based large language models in handling compositional tasks. It presents a thorough empirical study on three different types of tasks—multi-digit multiplication, logic grid puzzles, and a classic dynamic programming problem—to assess how these models perform in solving problems that require multi-step reasoning. The work introduces computation graphs as a novel metric for quantifying compositional complexity, and finds that these models often resort to mimicking patterns seen during training rather than developing genuine problem-solving capabilities.

The reviewers were particularly impressed by the extensive experiments and deep analysis presented in the paper. They found the idea of using computation graphs to measure compositional complexity inspiring and lauded the paper's comprehensive and detailed evaluation. The tasks designed for this study were considered valuable for future research on compositional reasoning, and the categories of errors identified were noted to be intriguing. Nevertheless, some reviewers pointed out that the empirical findings could be perceived as trivial, and the conclusions might be overgeneralized. Additionally, the theoretical justification was seen as somewhat oversimplified. However, these weaknesses do not significantly detract from the paper’s overall contribution and could be addressed in revisions. Hence, the paper receives a strong accept recommendation.